# Direct alkylation of *N,N*-dialkyl benzamides with methyl sulfides under transition metal-free conditions

Can-Can Bao[1], Hui-Zhen Du[1], Yan-Long Luo[1] & Bing-Tao Guan 👤 [2✉]

Amides are a fundamental and widespread functional group, and are usually considered as poor electrophiles owing to resonance stabilization of the amide bond. Various approaches have been developed to address challenges in amide transformations. Nonetheless, most methods use activated amides, organometallic reagents or transition metal catalysts. Here, we report the direct alkylation of *N,N*-dialkyl benzamides with methyl sulfides promoted by the readily available base LDA (lithium diisopropylamide). This approach successfully achieves an efficient and selective synthesis of α-sulfenylated ketones without using transition-metal catalysts or organometallic reagents. Preliminary mechanism studies reveal that the deprotonative aroylation of methyl sulfides is promoted by the directed *ortho*-lithiation of the tertiary benzamide with LDA.

---

[1] College of Chemistry, Nankai University, Tianjin, China. [2] Department of Chemistry, Fudan University, Shanghai, China. ✉email: bguan@fudan.edu.cn

Amide as one of the most fundamental structural unites exists widely and performs significantly in various proteins, natural products, pharmaceuticals, and synthetic materials[1–4]. The stability and planarity of the amide group originating from the resonance stability guarantees for their particular functions. Yet simultaneously, the decreased electrophilicity of the carbonyl group and the enhanced C–N bond energy become the main obstacle for the transformations of the inert amide group (Fig. 1a). The strongly nucleophilic organolithium and organomagnesium reagents could undergo the direct nucleophilic acyl substitution reaction with some amides. However, the reactions must be carried out under harsh conditions with a precise amount of the organometallic reagent to prevent the possible side reactions, for instance, reduction, deprotonation, and over addition reactions[5–7]. Therefore, the selective ketone synthesis from amides under mild conditions here comes as another great challenge. Thus, various activated amides have been particularly designed for the nucleophilic acyl substitution reactions (Fig. 1b)[8–12]. As early as 1981, Weinreb reported a clean and effective acylation of organolithium and organomagnesium reagents by using N-methoxy-N-methyl amides[13,14]. Moreover, N-Boc amides[15–18], N-acylpyrroles[19,20], and structurally distorted amides[21–26] have also been developed as the direct acylating reagents for the strong nucleophilic organometallic ketone synthesis (Fig. 1b1). Very recently, a fast and general ketone synthesis from amides and organolithium compounds was achieved by the stabilization of tetrahedral intermediate in CPME solution[27]. In 2015, Garg and co-workers reported the nickel-catalyzed esterification of amides, making a great breakthrough in amide transformations[28]. Soon later, Zou, Szostak, Garg, and Huang further reported the cross-coupling reactions of amides with organoboron or organozinc reagents by using palladium or nickel catalysts[29–34]. These works promptly attracted widespread attention and aroused explosive developments in the field of amides transformations (Fig. 1b2). Despite that, the amides scope was mostly limited in the electron-deficient amides and structurally distorted amides. For simple N-alkyl amides, a powerful

approach developed recently by Charette and Huang is the pre-activation of amides with Tf$_2$O and the following substitution reaction with organometallic reagents (Fig. 1c)[35–37]. Huang and his co-workers further found that the nitrilium ion intermediates generated from secondary amides and Tf$_2$O were electrophilic enough to react with arenes and alkenes to afford ketones[38–45]. Very recently, Liu and co-workers reported the chemodivergent transformation of various amides by using 1,1-diborylalkanes as pro-nucleophiles and alkyl lithium reagents as Lewis bases[46]. Despite these positive efforts and advances, the amide transformations developed so far still suffered from the use of organometallic reagents, pre-prepared either from organic halogens or with strong bases. Herein, we reported the direct alkylation of N,N-dialkyl benzamides with methyl sulfides under transition-metal-free conditions (Fig. 1d)[47–52].

Sulfide motif exists widely and plays crucial roles in many bioactive compounds, natural products, organocatalysts, and functional materials[53–57]. The deprotonative functionalization of sulfides represents one of the most frequently used strategies for sulfides synthesis. For the polarizability effect and hyperconjugation effect from the sulfur atom, the α-H of sulfides is usually considered as a weak acid (thioanisole: p$K_a$ = 38.3 in THF)[58–61]. (http://ibond.nankai.edu.cn/) A strong base was always necessary to deprotonate the sulfide to afford a carbanion intermediate, which could then undergo nucleophilic addition or substitution reaction with various electrophiles[62–70]. In 2018, Hou and co-workers reported the selective C−H alkylation of methyl sulfides with alkenes by using a cationic scandium catalyst, making a significant breakthrough in the α-functionalization of sulfides[71]. They further achieved the C−H silylation of methyl sulfides with hydrosilanes by a yttrium metallocene catalyst[72]. We earlier found that some relatively weak base catalysts could not undergo the complete deprotonation of a weakly acidic C−H bond but form a deprotonative equilibrium[73–78]. Thanks to the deprotonative equilibrium, the in-situ formed carbanion intermediate was reactive but in a low concentration, helping to avoid side reactions and achieving the reaction selectively. In particular, we recently

**Fig. 1 Amides transformations. a** Challenges in amide transformation. **b** Activated amides. **c** Amides activation. **d** This work: Alkylation of N, N-dialkyl benzamides with methyl sulfides.

discovered the benzylic aroylation of toluene with tertiary benzamides promoted by directed *ortho*-lithiation of benzamides with LDA[79]. Here we show that *N, N*-dialkyl benzamides can be directly alkylated with methyl sulfides using LDA as the base.

## Results and discussion

**Reaction discovery**. We first conducted our attempt in the reaction between *N, N*-diisopropyl benzamide, and thioanisole with several bases (1.2 equiv.) in THF at 60 °C (Table 1). Alkali bis(trimethylsilyl)amides failed to start the nucleophilic acyl substitution reaction (entries 1 and 2). LDA and LiTMP, as expected, smoothly drove on the reaction and selectively afforded the desired substituted product α-(phenylthio)acetophenone (**3aa**) in good yields (entries 3 and 4). It is worth noting that LDA and LiTMP generally could not deprotonate thioanisole for their relatively weak basicity. However, the direct deprotonative benzoylation of thioanisole was simply achieved with these bases. Strong basic *n*-BuLi and TMSCH$_2$Li in the reaction completely consumed the amide but gave the ketone products in low yields of 21 and 58% (entries 5 and 6). The direct substitution reaction between benzamide and butyl lithium took place to give butyl phenyl ketone (30% yield) and 5-phenylnonan-5-ol (14% yield) as by-products (entry 5). These results suggested that the strongly basic but slim butyl lithium worked not only as a base but also as a nucleophile, leading to side reactions. LDA, a readily available base, displayed less nucleophilicity and thus behaved better than the strong bases. We found the reaction to be a stoichiometric reaction (see Supplementary Table 1 in Supplementary Methods) and finally revealed the optimized conditions: lower loading of thioanisole (1.4 equiv.) and LDA (1.1 equiv.) and lower temperature (40 °C, entry 7; for more condition screening, see

### Table 1 Alkylation of *N, N*-diisopropylbenzamide with Thioanisole[a].

| entry | base | x | 1a Conv. (%)[b] | 3aa yield (%)[b] |
|---|---|---|---|---|
| 1 | LiHMDS | 1.2 | 8 | <5 |
| 2 | KHMDS | 1.2 | 6 | <5 |
| 3 | LiTMP | 1.2 | >95 | 76 |
| 4 | LDA | 1.2 | >95 | 94 |
| 5 | *n*-BuLi[c] | 1.2 | >95 | 21[d] |
| 6 | LiCH$_2$TMS | 1.2 | >95 | 58 |
| 7[e] | LDA | 1.1 | >95 | 98 (93) |

[a]Reaction Conditions: benzamide **1a** (0.5 mmol), thioanisole **2a** (1.0 mmol), base, THF (1.0 mL), 60 °C, 24 h. HMDS: bis(trimethylsilyl)amide; LDA: lithium diisopropylamide; LiTMP: lithium 2,2,6,6-tetramethylpiperidide.
[b]NMR yields and conversions with 2-methyloxynaphthalene as an internal standard, isolated yield in parenthesis.
[c]1.6 M in hexane.
[d]Butyl phenyl ketone (30% yield) and 5-phenylnonan-5-ol (14% yield) were obtained as by-products.
[e]LDA (0.55 mmol), **2a** (0.7 mmol), 40 °C.

**Fig. 2 Scope of Methyl Sulfides.** Conditions: for thioanisoles, benzamide **1a** (0.5 mmol), **2** (0.7 mmol), LDA (0.55 mmol), THF (1 mL), 40 °C, 24 h; for alkyl methyl sulfides, benzamide **1a** (0.5 mmol), **4** (1.25 mmol), LDA (0.75 mmol), THF (1 mL), 40 °C, 24 h. [a]Di-benzoylation product **3ana** was obtained as by-product (26% yield). [b]Benzamide **1a** (0.9 mmol), 1,4-bis(methylthio)benzene **2n** (0.3 mmol), LDA (0.66 mmol), yield based on **2n**. [c]60 °C.

**Fig. 3 Scope of Benzamides.** Conditions: for thioanisole, benzamide **1** (0.5 mmol), **2a** (0.7 mmol), LDA (0.55 mmol), THF (1 mL), 40 °C, 24 h; for isopropyl methyl sulfide, benzamide **1** (0.5 mmol), **4a** (1.25 mmol), LDA (0.75 mmol), THF (1 mL), 40 °C, 24 h. [a]60 °C.

Supplementary Table 2 and 3). Several other tertiary benzamides were also tested in this reaction. The steric bulky N, N-dicyclo-hexyl benzamide smoothly underwent the reaction and afforded a comparable yield of 95%, while the less steric benzamides afforded lower yields (for additional substrates, see Supplementary Table 4). In addition, a high yield of 92% for the gram-scale reaction positively demonstrated the reliability of this protocol (1.1 g, see 4.3 in Supplementary Information).

With this optimized condition in hand, we carried the reactions between N, N-diisopropyl benzamide and various aryl methyl sulfides (Fig. 2). 4-Ethylthio thioanisole smoothly underwent the reaction with benzamide and afforded the methyl benzoylation product in a high yield of 96% (**3ab**), but leaving the methylene group intact. It is worth noting that the reaction displayed excellent selectivity of the thiomethyl group over the thiomethylene group and the *ortho* phenyl C—H bond. 4-N, N-Dimethyl thioanisole, 4-isopropoxyl thioanisole, and methoxy thioanisoles were allowed to react with benzamide and the benzoylation products were obtained in good to high yields (**3ac–ag**). The slightly lower yield of 2-methoxythioanisole relative to its 4-methoxy and 3-methoxy isomers suggested that the coordination or steric repulsion from the 2-methoxyl group made the thioanisole a bit sluggish in the substitution reaction. Phenyl, alkenyl, alkyl-substituted thioanisoles, and methyl naphthyl sulfides are also suitable substrates to give the substitution products in good to high yields (**3ah–am**). For 1,4-bis(methylthio)benzene, we could control the ratio of the

reactants to selectively get the mono-benzoylation product (**3an**) and the di-benzoylation product (**3ana**). Some methyl sulfides bearing heteroaromatic rings were subjected in the reaction with benzamide but failed to give the desired products (see Supplementary Table 4). It was possible that the strong coordinating groups would inhibit the deprotonation and substitution reaction.

To explore the generality of this reaction, we further carried out the reactions of isopropyl methyl sulfide and got the expected benzamide substitution product **5aa** in a high yield of 85% under a slightly enhanced condition (2.5 equiv. of sulfide and 1.5 equiv. of LDA; for detail condition screening, see Supplementary Table 5). We then explored the nucleophilic acyl substitution reactions of benzamide with a serious of alkyl methyl sulfides under this condition (Fig. 2). The reaction displays excellent selectivity on the methyl group over the methylene and methine groups. Interestingly, we found that the methyl sulfides bearing isopropyl, *tert*-butyl, cyclohexyl, and cycloheptyl groups gave better yields than the ones with other alkyl groups (**5aa–ad** vs. **5ae–al**). It seems that the steric hindrance from the other side of the sulfide could increase its reactivity. Trimethylsilyl, tertiary amine, ether, and silyl ether groups were well tolerated in the reaction, giving the corresponding α-sulfenylated ketones in moderate to good yields (**5am–ap**).

We further investigated the scope of substituted benzamides with either thioanisole or isopropyl methyl sulfide (Fig. 3). 4-*Tert*-butyl benzamide, 4-isopropyl benzamide, and 4-cyclohexyl

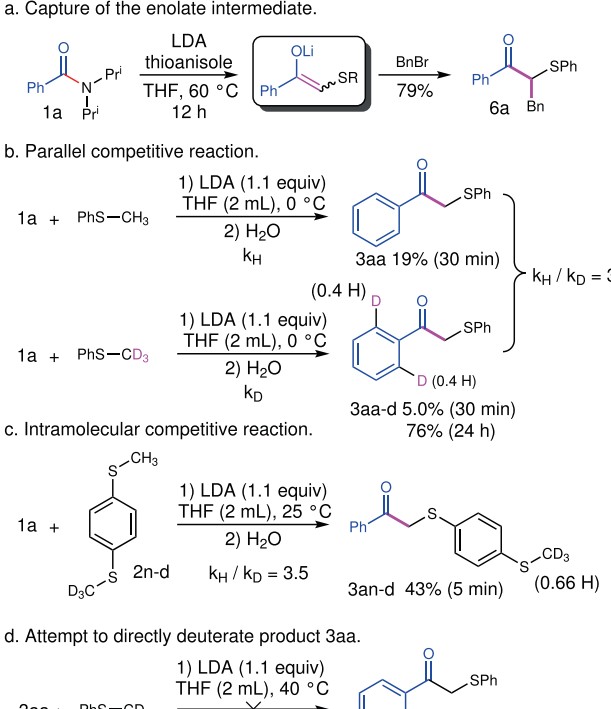

**Fig. 4 Control Experiments. a** Capture of the enolate intermediate. **b** Parallel competitive reaction. **c** Intramolecular competitive reaction. **d** Attempt to directly deuterate product **3aa**.

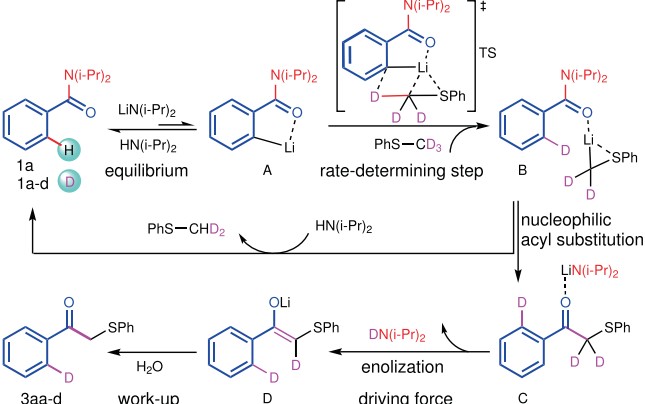

**Fig. 5 A plausible reaction pathway.** Intermediate **A** possibly exists in its dimer structure.

benzamide successfully reacted with thioanisole and gave the substitution products in high yields (**3ba–da**). 4-Butyl benzamide, however, yielded the corresponding products in lower yields (**3ea** 74%), which could be attributed to the side reactions of an alkyl group. Phenyl, *N, N*-dimethyl, phenoxyl, and methoxy benzamides were also allowed to react with thioanisole and corresponding α-sulfenylated ketone products were obtained in good to high yields (**3fa–la**). 2-Naphthamide gave the desired product in 83% yield (**3ma**), while 1-naphthamide afforded a low yield of 14%. 2-Phenyl benzamide failed to react with thioanisole. It seems that the steric hindrance from benzamides would greatly inhibit the substitution reaction. Isopropyl methyl sulfide smoothly reacted with the benzamides but gave the products in a bit lower yields than those of thioanisole (**5ba–ma**).

**Mechanistic studies**. We are quite interested in the deprotonation of thioanisole with LDA, which is the key step in alkylation of the tertiary benzamide. LDA failed to deprotonate thioanisole for its weak acidity. The tertiary benzamide must play an essential role in the deprotonation process. The nucleophilic acyl substitution reaction of benzamide with alkyl lithium would generate ketone intermediate and a new LDA. The enolization of the ketone with LDA could take place easily and provide the driving force. The direct quench of the reaction with benzyl bromide afforded the further α-alkylation product **6a** in a good yield of 79% (Fig. 4a), demonstrating the enolate intermediate.

The reaction of benzamide with thioanisole and thioanisole-*d₃* under exactly the same conditions revealed a primary kinetic isotope effect (KIE = 3.2, Fig. 4b). The intramolecular competitive reaction between methyl C–H bond and C–D bond resulted in a similar KIE value (KIE = 3.5, Fig. 4c). The kinetic isotope effects suggested that the cleavage of the methyl C–H bond of thioanisole could be involved in the rate-determining step. It is

interesting that the reaction with thioanisole-*d₃* gave the product with 60% deuterium incorporation on the *ortho*-position of the carbonyl group (**3aa**-*d*), suggesting the participation of this C–H bond. The mix of the ketone product **3aa** and thioanisole-*d₃* under the same condition failed to afford any deuterium scrambling product (Fig. 4d), which excluded the deuteration of the product. A reasonable explanation is that the *ortho*-lithiation intermediate underwent the deuteration reaction with thioanisole-*d₃* first and then the alkylation of the tertiary benzamide took place. We further measured the initial rate and found it to be first-order in the amide, first-order in LDA, first-order in thioanisole, and minus one order in HDA (*N, N*-diisopropyl amine), suggesting an equilibrium and a following rate-determining process (see 2.8 in Supplementary Information).

Based on the results described above and our recent benzylic aroylation of toluene[79], we proposed a possible process for the substitution of benzamide **1a** with thioanisole-*d₃* as shown in Fig. 5. The deprotonation equilibrium between benzamide (p$K_a$ = 37.8 in THF) and LDA (HDA: p$K_a$ = 35.7 in THF)[60] provided an *ortho*-lithiation intermediate **A**[80–82] in a low concentration. This *ortho*-lithiation intermediate coordinated with thioanisole-*d₃* and underwent the σ-bond metathesis reaction via a four-membered ring transition state (**TS**), affording an alkyl lithium intermediate coordinating with the deuterated benzamide (**B**). The following nucleophilic acyl substitution reaction of the amide with the alkyl lithium intermediate generated the ketone intermediate (**C**) and a new LDA. The easy enolization afforded the enolate intermediate **D**, providing the driving force of the reaction. The final ketone product was then obtained after the quench of the enolate intermediate **D** with water. The protonation of intermediate **A** and **B** with diisopropylamine and its deuterated isomer quickly achieved the H/D exchange between benzamide and thioanisole, resulting in the product bearing deuterium-diluted *ortho*-C–H bond.

## Conclusion

In summary, we have developed a direct alkylation reaction of *N, N*-dialkyl benzamides with methyl sulfides, which provides a direct and efficient approach for the synthesis of α-sulfenylated ketones. This reaction features selective methyl C–H bond deprotonative functionalization of methyl sulfides with LDA, a relatively weaker base than lithium alkyls. Preliminary mechanism studies revealed that the *ortho*-lithiation of the tertiary benzamide promoted the deprotonation of methyl sulfides and triggered the nucleophilic acyl substitution reaction of the benzamide. Distinct from the amide activation strategies by either

enhancing the amide reactivity or stabilizing the tetrahedral intermediates, the low concentration of the reactive carbanion intermediates arising from the deprotonation equilibrium of methyl sulfides with LDA could be the key factor for restraining side reactions and improving the selectivity.

## Methods

**A general procedure for the alkylation of benzamide with thioanisoles**. To a 25 mL Schlenk tube equipped with a Teflon septum and magnetic stir bar were added benzamides **1** (0.50 mmol, 1.0 equiv.), thioanisoles **2** (0.70 mmol, 1.4 equiv.), THF (1.0 mL) and LDA (58.9 mg, 0.55 mmol, 1.1 equiv., solid). The tube was sealed and stirred at 40 °C for 24 h. The mixture was then cooled to room temperature, quenched by adding five drops of $H_2O$ and then diluted by ethyl acetate (EtOAc, 30 mL). The obtained solution was dried over anhydrous $Na_2SO_4$. After filtration and concentration by rotary evaporation, the residue was purified by silica gel column chromatography to afford the desired products **3**.

**A general procedure for the alkylation of benzamide with alkyl methyl sulfides**. To a 25 mL Schlenk tube equipped with a Teflon septum and magnetic stir bar were added benzamides **1** (0.50 mmol, 1.0 equiv.), alkyl methyl sulfides **4** (1.25 mmol, 2.5 equiv.), THF (1.0 mL) and LDA (80.3 mg, 0.75 mmol, 1.5 equiv., solid). The tube was sealed and stirred at 40 °C for 24 h. The mixture was then cooled to room temperature, quenched by adding five drops of $H_2O$ and then diluted by ethyl acetate (EtOAc, 30 mL). The obtained solution was dried over anhydrous $Na_2SO_4$. After filtration and concentration by rotary evaporation, the residue was purified by silica gel column chromatography to afford the desired products **5**.

## Data availability

For additional condition screening, see Supplementary Tables. 1–6; for kinetic control experiments, see Supplementary Fig. 1–11; for [1]H and [13]C spectra of compound **3** and **5**, see Supplementary Fig. 12–123.

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

## Acknowledgements

We gratefully acknowledge the support from Fudan University. We thank Prof. Zhang-Jie Shi (Fudan Univ.), Prof. Bi-Jie Li (Tsinghua Univ.), and Prof. Qian Peng (Nankai Univ.) for their constructive discussions. This project was supported by the National Natural Science Foundation of Tianjin (No. 19JCYBJC20100).

## Author contributions

C.-C. B. and B.-T. G. designed the research and co-wrote the article. C.-C. B. performed most of the reaction optimization, substrate screening, and mechanism experiments. H.-Z. D. and Y.-L. L. helped accomplish the substrate screening experiments, conducted some experiments, and collect data. B.-T. G. directed the project.

## Competing interests

The authors declare no competing interests.
