## [Peer Review File · Communications Chemistry]

Reviewers' comments:

Reviewer #1 (Remarks to the Author):

In this manuscript, the authors reported a method for the synthesis of α -sulfenylated aromatic carbonyl compounds from *N,N*-diisopropyl arylamides and methyl sulfides. The method involves the addition of carbanion of methyl sulfides generated in situ from benzamide and derivatives and LDA. With the postulated transition state and chelating intermediate B, the method is mechanically interesting. The scope to of the aryl group in benzamide derivatives and of methyl sulfides were examined. However, that of *N*-substituents was not mentioned. Thus the method is narrow in scope, restricting to *N,N*-diisopropyl arylamides and methyl sulfides as the starting materials and α -sulfenylated aromatic carbonyl compounds as the products. Even for the same class of products, Pace's method (EJOC 2018, 2466, cited as ref. 77) is larger in scope. Moreover, there are many problems in terms of concepts and cited references.

1. In the title and conclusion section, by what means "Unactivated Benzamides"?
 2. The authors seems to lack a comprehensive understanding of the current status of the amide chemistry. Consequently, the first and second sentences and other sentences in the abstract are inappropriate. A fast and general route to ketones from amides and organolithium compounds under aerobic conditions appeared recently: Chem.–Eur. J. 2021, 27, 2868-2874. "Without using transition-metal catalysts, organometallic reagents or strong bases", the coupling of amides with alkenes to give ketones or enones are known (Chin. J. Chem. 2019, 37, 887; Chin. J. Chem. 2019, 37, 811; Chin. J. Chem. 2019, 37, 315 and cited references). On the other hand, the preparations of ketones from *N,N*-dimethylamides and *N,N*-diethylamides are known: Synthesis 1984, 228 (cited as ref. 6) and J. Org. Chem. 1986, 51, 3566.
 3. Another key issue is about the definition of amides. Fig. 1, a represents the widely accepted definition of amides (herein, it is better changing Ph to R). However, in the text including Fig. 1, b (middle), and many references, different kind of diacyl, and triacylamines were also called amides. This is problematic, because diacyl, and triacylamines have not resonance structures shown in Fig. 1, a, namely, the (first) carbonyl group in such so called "amides" is as reactive as a ketone carbonyl. As such the cited references 27-52 and related text should be deleted.
 4. A reference on the ortho-metalation with LDA should be cited: Angew. Chem. Int. Ed. 2019, 58, 7313.
 5. In the SI general protocol, please indicate the LDA used, solid or a solution?
 6. In the SI, for the ¹³C NMR data, the peak numbers didn't consist with the structure.
- In summary, the manuscript requires major alternations before a recommendation could be made.

Reviewer #2 (Remarks to the Author):

Recommendation: Accepted after minor revision.

Comments:

In this manuscript, Bao et al reported a direct alkylation of benzamides by methyl sulfides in the presence of LDA as the base source. In general, the manuscript was well written and also included deep discussion in mechanics studies. However, to provide a more concrete and reliable discussion regarding the proposed methodology, several discussions and experiments are highly demanded.

Therefore, this manuscript could be accepted after tackling with minor comments below:

1. The authors are supposed to remove any ambiguous description, such as “relative weak base” for highlighting the LDA, which is commonly regarded as strong base with bulky structure.
2. In terms of benzamides, dimethyl groups on N are more generally utilized to represent the unactuated benzamide. However, the authors only exemplified the i-Pr. Accordingly, i-Pr is more likely to facilitate the reaction. The authors are recommended to give both experiments and discussions regarding the i-Pr.
3. In the mechanistic discussion, the authors concluded that from A to B is the rate-determining step without related evidence. The authors are recommended to provide the DFT calculation on the proposed transition state. Or at least, the provable literatures should be referred.

Reviewer #3 (Remarks to the Author):

see review attachment

This paper by Guan *et al.* describes the use of a stoichiometric amount of LDA to trigger a C–C bond formation between two challenging substrates, tertiary benzamides (not very electrophilic) and methyl sulfides (not very acidic), *via* formal C–N and C–H bond activation, respectively; to form the corresponding α -thioalkoxy-functionalized ketones *via* a nucleophilic acyl substitution reaction. This very nice transformation is interesting (although *N*-containing products such as enamines would be more synthetically valuable), and this approach seemingly avoids the formation of side-products and the use of transition metal and alkyl lithium reagents. The optimization and control experiments conducted are sensible, and the mechanistic study overall supports the proposed reaction pathway: an initial Snieckus-type directed *ortho*-metalation (DoM) of the aromatic benzamide followed by an Ar–Li-assisted α -lithiation of the methyl sulfide; a stoichiometric amount of LDA is required because a ketone enolate being formed as a stoichiometric product; prior to hydrolysis. Generally, sufficient references have been included (a few exceptions). This study is sufficiently distinct from relevant earlier work published by others and the lead-author of the present work, Professor Guan (ref 83). Therefore, science-wise the publication of this study can be overall (highly) recommended – provided that several points are addressed (see below).

Manuscript (see also annotated scan)

The used English is of rather average quality; the editorial staff must take care of this issue.

The following points are to be addressed:

- p1/3/7: metal amides should be considered as strong Brønsted bases; if the base strength is mentioned it should be clearly put in relation to the pK_a value of the corresponding conjugate acid – also, when pK_a values of the reagents are compared it is not useful at all to consider two different solvents (DMSO/THF); as such data are not comparable. Also, when using Lewis acidic entities (here Li^+) complexation to basic sites in the substrates can significantly increase the relevant C–H bond acidity, which should be taken into consideration.
- p2: there is a confusing statement of “decreased nucleophilicity” regarding amides; explain properly or change to “electrophilicity” (which would make sense).
- p2: better to add a suitable ref when mentioning the potential side-reactions.
- p2: “ketamine” is the wrong term; it must be “ketiminium”.
- p2 (F1): R' must be defined for the three distinct reaction pathways (b1, b2, c).
- p3: add the term “Lewis”, otherwise misleading.
- p3: the relevant data from Liu's paper (ref 56) should be displayed in F1.
- p3: the stabilization of the α -anion of *S*-containing compounds (hyperconjugation) must be added (plus a suitable ref).
- p3: self-citation (refs 81–83) is OK but then relevant other studies (use of alkali metal amides in C–H bond activation; all equilibria) must be included as well ... e.g. Kobayashi, Schneider, Walsh, ...
- p4/6: the deprotonation/quenching of thioanisol with LDA and LTMP –in the absence of the benzamide– must be carried under the reaction conditions of T1 (e.g. THF, 60 °C; then TMSCl) before stating it is unsuccessful (due to pK_a values that are not comparable anyway; as reported in different solvents). Also, if the bis-*ortho*-methylated benzamide (not a Brønsted acid) is used as a Lewis basic additive (to enhance the Brønsted basicity of LDA), what would be the outcome when mixing thioanisol and LDA ?
- p4: there should be also “steric hindrance” included in the base/nucleophile discussion (i.e.,

- bulky amides vs. slim alkyl reagents); it is completely missing.
- p4/6: are tertiary alcohols obtained at any stage in any of the conducted experiments (particularly for the alkyl lithium reagents and/or in the final control experiment)?
 - p4: regarding the amount of LDA used, refer to the SI (TS-1).
 - P4: in a similar context, clarify the following points:
 - What is the outcome if Me-SPh (\rightarrow stoichiometric LDA; secondary center created) is replaced:
 - (a) by Et-SPh or TMSCH₂-SPh (\rightarrow stoichiometric LDA; tertiary center created) ?
 - (b) by ⁱPr-SPh (\rightarrow potentially *catalytic* LDA; quaternary center created) ?
 - [may require more harsh conditions; dioxane, 80+ °C]
 - p4/5 (T2/3): regarding the scope (which is in principle very good), what is the outcome in case of methyl sulfides or tertiary amides bearing an Ar group with an EWG (there is not any example in both Tables) ?
 - p4/5 (T2/3): adjust structural display of “Bu” and “Pr” in the Tables.
 - Amide scope: exclusive use of aromatic tertiary *N*-isopropyl benzamides (= drawback of this methodology): what is the scope of the NR₂ portion ? For instance, how about NEt₂, N(TMS)₂, N(pyrrolidinyl) ?
 - p6/7 (F2/3): the control experiments conducted are reasonable (particularly the deuterium-labelling ones) and support the proposed mechanistic pathway. It would be good though to try detecting postulated reaction intermediates (e.g. **A** and **D**) by mild HRMS methods (obviously in the absence of the sulfide and water, respectively).
 - Sulfide deprotonation: include a suitable ref of a relevant Ar-Li-assisted “ σ -bond metathesis-type” C-H bond metalation.

SI (see also annotated scan)

Generally, the SI is OK, but the following points are to be addressed:

- TOC: might be more reader-friendly to move “Control Experiments” forward (S-1).
- General Information: clarify the LDA confusion; add information on mp measurement (S-2).
- Reaction Optimization: clarify the LDA point; comment on the possibility of tertiary alcohol formation through nucleophilic addition in case of alkyl lithium species (S-3).
- General Procedures: better to add “m” [in mg] of methyl sulfide reagents, not just show “V” [in μ L] (S-4, S-26).
- Analytical Data: there seem to be accuracy issues with NMR data (almost on every single page ... S-6 ~ S-26); must be addressed; use the term “colorless” (not “white”) solid; some mp’s look strange in terms of significant numbers (S-10, S-11, S-22).
- Control experiments: clarify two points (S-28, S-30).
- NMR Spectra: these display indeed sufficient purity of the isolated products; one exception though: based on the ¹H NMR chart, product **3ea** seems not pure (1.2~1.3 ppm); better to re-purify and up-date yield.

Direct Alkylation of Unactivated Benzamides with Methyl Sulfides by Breaking Both C-N and C-H Bonds under Transition-Metal-Free Condition

Can-Can Bao¹, Hui-Zhen Du¹, Yan-Long Luo¹ and Bing-Tao Guan^{2*}

¹ College of Chemistry, Nankai University, Tianjin 300071, China

² Department of chemistry, Fudan University, 2005 Songhu Road, Shanghai 200438, China

* E-mail: bguan@fudan.edu.cn

Amide, a fundamental and widespread functional group, is usually considered as a poor electrophile for the resonance stability of the amide bond. Various approaches have been developed to address the challenges in the amide transformations. Nonetheless, most had to use activated amides, organometallic reagents or transition metal catalysts. Herein, we reported the direct alkylation of unactivated tertiary benzamides with methyl sulfides promoted by LDA (Lithium diisopropylamide) as a relative weak base. This approach successfully achieved an efficient and selective synthesis of α -sulfenylated ketones without using transition-metal catalysts, organometallic reagents or strong bases. Preliminary mechanism studies revealed that the directed *ortho*-lithiation of a benzamide with LDA promoted the methyl deprotonation of sulfides, and thus triggered the nucleophilic acyl substitution reaction of the benzamides.

should be
considered a strong base

Amide as one of the most fundamental structural units exists widely and performs significantly in various proteins, natural products, pharmaceuticals, and synthetic materials.¹⁻⁴ The stability and planarity of the amide group originating from the resonance stability guarantees for their particular functions. Yet simultaneously, the decreased nucleophilicity of the carbonyl group and the enhanced C-N bond energy bring a significant challenge in synthetic chemistry for the transformations of the inert amide group (Fig. 1a). The strongly nucleophilic organolithium and organomagnesium reagents could undergo the direct nucleophilic acyl substitution reaction with some amides. However, the reactions must be carried out under harsh conditions with precise amount of the organometallic reagent to prevent the possible side reactions, for instance, reduction, deprotonation and overaddition reactions. Therefore, the selective ketone synthesis from amides under mild conditions here comes as another great challenge.^{5,6} To overcome these challenges, various activated amides were particularly designed for the nucleophilic acyl substitution reactions (Fig. 1b).⁷⁻¹¹ As early as 1981, Weinreb reported a clean and effective acylations of organolithium and organomagnesium reagents by using *N*-methoxy-*N*-methyl amides.^{12,13} Moreover, *N*-Boc amides¹⁴⁻¹⁷, *N*-acylpyrroles^{18,19} and structurally distorted

amides²⁰⁻²⁵ have also been developed as the direct acylating reagents for the strong nucleophilic organometallic ketone synthesis (Fig. 1b1). In 2015, Garg and co-workers reported the nickel-catalyzed esterification of amides, making a great breakthrough in amide transformations.²⁶ In the same year, Zou, Szostak and Garg independently reported the Suzuki coupling reactions of amides by using palladium or nickel catalysts.²⁷⁻²⁹ Soon later, Garg and Szostak further reported the nickel-catalyzed Negishi coupling of *N*-Boc benzamides.^{30,31} These works promptly attracted widespread attention and aroused explosive developments in the field of amides transformations (Fig. 1b2).³²⁻⁵² Despite that, the amides scope was mostly limited in the electron-deficient amides and structurally distorted amides. For simple *N*-alkyl amides, a powerful approach developed recently by Huang and Charette is the pre-activation of the amide with Tf₂O (trifluoromethanesulfonic anhydride) and the following substitution reaction with organometallic reagents (Fig. 1c).⁵³⁻⁵⁵ The key of this approach is the generation of a highly electrophilic O-triflyl imidate intermediate that can react readily with an organometallic reagent to afford a ketamine intermediate. The subsequent acidic hydrolysis of the ketamine intermediate smoothly and selectively release the ketone products. Very recently, Liu and co-workers reported the chemodivergent

a. Challenges in amide transformations

- decreased nucleophilicity
- enhanced C-N bond energy
- harsh conditions
- poor selectivity

b. Reactions of activated amides

c. Activation of amides

d. Alkylation of unactivated benzamides with methyl sulfides (*this work*):

misleading

?

ref

2

minimum

define R' for the different pathways

Fig. 1 Amides transformations. a Challenges in amide transformation. b Activated amides. c Amides activation. d This work: Alkylation of unactivated amide with methyl sulfides.

Table 1. Alkylation of Benzamide with Thioanisole.^a

entry	base	x	1a Conv. (%) ^{1a}	3aa yield (%) ^b
1	LiHMDS	1.2	8	< 5
2	KHMDS	1.2	6	< 5
3	LiTMP	1.2	> 95	76
4	LDA	1.2	> 95	94
5	n -BuLi ^c	1.2	> 95	24 ^d
6	LiCH ₂ TMS	1.2	> 95	58
7 ^e	LDA	1.1	> 95	98 (93)

^aReaction Conditions: benzamide **1a** (0.5 mmol), thioanisole **2a** (1.0 mmol), base, THF (1.0 mL), 60 °C, 24 h. HMDS: bis(trimethylsilyl)amide; LDA: lithium diisopropylamide; LiTMP: lithium 2,2,6,6-tetramethylpiperidide.

^bNMR yields and conversions with 2-methyloxynaphthalene as an internal standard, isolated yield in parenthesis.

^c1.6 M in hexane.

^dButyl phenyl ketone was obtained as by-product (31% yield).

^eLDA (0.55 mmol), **2a** (0.7 mmol), 40 °C.

transformation of various amides by using Lewis 1,1-diborylalkanes as pro-nucleophiles and alkyl lithium reagents as bases.⁵⁶ Despite these positive efforts and advances, the amide transformations developed so far still faced the great challenge of using organometallic reagents, which were always synthesized either from organic halogens or via deprotonation with strong bases. Herein, we reported the direct alkylation of unactivated tertiary benzamides with methyl sulfides under transition-metal-free condition, therefore providing an ideal pathway for the synthesis of α -sulfenylated ketones (Fig. 1d).⁵⁷⁻⁸³

Sulfide motif exists widely and plays crucial roles in many bioactive compounds, natural products, organocatalysts and functional materials.⁶⁴⁻⁶⁸ The deprotonative functionalization of sulfides represents one of the most frequently used strategy for sulfides synthesis. For the limited electron withdrawing effect from the sulfur atom, the α -H of simple methyl sulfides is usually considered as a weak acid (dimethyl sulfide: $pK_a = 45$ in DMSO; thioanisole: $pK_a = 42$ in DMSO).⁶⁹ Therefore, a strong base was always necessary to deprotonate the sulfide to afford a carbanion intermediate, which could then undergo nucleophilic

addition or substitution reaction with various electrophiles.⁷⁰⁻⁷⁸ In 2018, Hou and co-workers reported the selective C-H alkylation of methyl sulfides with alkenes by using a cationic scandium catalyst, making a significant breakthrough in the α -functionalization of sulfides.⁷⁹ Soon later, they further achieved the C-H silylation of methyl sulfides with hydrosilanes by an yttrium metallocene catalyst.⁸⁰ In our former works about base-catalyzed C-H bond alkylation reactions, we found that some relative weak base catalysts could not undergo the complete deprotonation of a weakly acidic C-H bond but form a deprotonative equilibration.⁸¹⁻⁸³ Thanks to the deprotonative equilibration, the reactive carbanion intermediate formed in a low concentration, which helped to avoid side reactions and achieve the reaction selectively. This observation inspired us to investigate the nucleophilic acyl substitution reaction of benzamide with sulfides under a relative weak base condition.

Results

Reaction discovery. We first examined several bases (1.2 equiv.) in the reaction between *N,N*-diisopropyl benzamide and thioanisole in THF at 60 °C (Table 1). Alkali bis(trimethylsilyl)amides failed to start the

mention hyperconjugation (σ_{C-S}^*) to stabilize α -anion

e.g. Kobayashi
Schneider
Walsh ...

self-citation OK but other authors must be then cited as well

nucleophilic acyl substitution reaction (entries 1 and 2). LDA and LiTMP, as expected, smoothly drove on the reaction and selectively afforded the desired substituted product α -(phenylthio)acetophenone (**3aa**) in good **Table 2. Scope of Methyl Sulfides.^a**

yields (entries 3 and 4). It is worth noting that LDA and LiTMP generally could not deprotonate thioanisole for their

^aConditions: for thioanisoles, benzamide **1a** (0.5 mmol), **2** (0.7 mmol), LDA (0.55 mmol), THF (1 mL), 40 °C, 24 h; for isopropyl methyl sulfide, benzamide **1a** (0.5 mmol), **4** (1.25 mmol), LDA (0.75 mmol), THF (1 mL), 40 °C, 24 h. ^bDi-benzoylation product **3ana** was obtained as by-product (26% yield). ^cBenzamide **1a** (0.9 mmol), 1,4-bis(methylthio)benzene **2n** (0.3 mmol), LDA (0.66 mmol), yield based on **2n**. ^d60 °C.

relative weak basicity. Based on our knowledge, we were the first to achieve the direct alkylation of unactivated tertiary benzamides with thioanisole simply by using a relative weak base. Even more basic alkyl lithium reagents, *n*-BuLi and TMSCH₂Li, were also subjected into the reaction. The benzamide completely consumed, however, the ketone products were obtained in low yields of 24% and 58% (entries 5 and 6). The direct substitution reaction between benzamide and butyl lithium took place to give butyl phenyl ketone as a by-product (entry 5, 31% yield). These results suggested

that the relative weak base LDA displayed much better than the strong bases. We further tested the reaction with different amount of LDA and found it to be a stoichiometric reaction. We finally could complete the reaction with lower loading of thioanisole (1.4 equiv.) and LDA (1.1 equiv.) under lower temperature (40 °C) and got an even better yield (98%, entry 7). In addition, we carried a gram-scale reaction with the production yield at 92%, which positively demonstrated the reliability of this protocol (1.1 g, for more condition screening, see Supporting Information).

formation of tertiary alcohols?

was this exp performed?
(e.g. quench with TMSCl)
→ Li⁺ as Lewis acid can decrease pKa of α -H

key may be steric hindrance?

refer to SI

With this optimized condition in hand, we carried the reactions between *N,N*-diisopropyl benzamide and various aryl methyl sulfides (Table 2). 4-Ethylthioanisole smoothly underwent the reaction with **Table 3**. Scope of Benzamides.^a

benzamide and afforded the methyl benzoylation product in a high yield of 96% (**3ab**), but leaving the methylene group intact. It is worth noticing that the reaction displayed excellent selectivity of the thiomethyl group

^aConditions: for thioanisole, benzamide **1** (0.5 mmol), **2a** (0.7 mmol), LDA (0.55 mmol), THF (1 mL), 40 °C, 24 h; for isopropyl methyl sulfide, benzamide **1** (0.5 mmol), **4a** (1.25 mmol), LDA (0.75 mmol), THF (1 mL), 40 °C, 24 h. ^b60 °C

over the thiomethylene group and the *ortho* phenyl C-H bond. 4-*N,N*-Dimethyl thioanisole, 4-isopropoxy thioanisole and methoxyl thioanisoles were allowed to react with benzamide and the benzoylation products were obtained in good to high yields (**3ad-3ag**). The slightly lower yield of *ortho*-methoxyl thioanisole relative to its *para*- and *meta*- isomers suggested that the coordination or steric repulsion from the *ortho* methoxyl group made the thioanisole a bit sluggish in the substitution reaction. Phenyl, alkenyl, alkyl substituted thioanisoles and methyl naphthyl sulfides are also suitable substrates to give the substitution products in

good to high yields (**3ah-3am**). For 1,4-bis(methylthio)benzene, we could control the ratio of the reactants to selectively get the mono-benzoylation product (**3an**) and the di-benzoylation product (**3ana**). Some methyl sulfides bearing heteroaromatic ring were subjected in the reaction with benzamide but failed to give the desired products (see Supporting Information). It was possible that the strong coordinating group on the sulfides would inhibit the deprotonation and substitution reaction. To explore the generality of this reaction, we further carried out the reactions of isopropyl methyl sulfide and got the expected

could be neglected

benzamide substitution product **5aa** in a high yield of 85% under a slightly enhanced condition (2.5 equiv. of sulfide and 1.5 equiv. of LDA; for detail condition screening, see supporting information). We then explored the nucleophilic acyl substitution reactions of benzamide with a series of alkyl methyl sulfides under this condition (table 2). The reaction displays excellent selectivity on the methyl group over the methylene and methine groups. Interestingly, we found that the methyl

Fig. 2 Control Experiments.

sulfides bearing isopropyl, *tert*-butyl, cyclohexyl and cycloheptyl groups gave better yields than the ones with other alkyl groups (**5aa-5ad** VS **5ae-5al**). It seems that the steric hindrance from the other side of the sulfide could increase its reactivity. Trimethylsilyl, tertiary amine, ether and silyl ether groups were well tolerated in the reaction, giving the corresponding α -sulfenylated ketones in moderate to good yields (**5an-5ao**).

We further investigated the scope of substituted benzamides with either thioanisole or isopropyl methyl sulfide (Table 3). 4-*tert*-Butyl benzamide, 4-isopropyl benzamide and 4-cyclohexyl benzamide successfully reacted with thioanisole and gave the substitution products in high yields (**3ba-3da**). 4-Butyl benzamide, however, yielded the corresponding products in lower yields (**3ea** 74%), which could be attributed to the side reaction of alkyl group towards benzamides. Phenyl,

N,N-dimethyl, phenoxy and methoxy benzamides were also suitable substrates to give the substitution products in good to high yields (**3fa-3la**). 2-Naphthamide reacted with thioanisole smoothly to afford the desired product in 83% yield (**3ma**). 1-Naphthamide, however, gave the product in a low yield of 8%. 2-Phenyl benzamide failed to react with thioanisole. This significant reactivity difference revealed that the steric hindrance from benzamides would greatly inhibit the substitution reaction. Fluorobenzamides, 3-furanyl, and 3-thienyl amide were also tested but failed to undergo the nucleophilic acyl substitution reaction. Isopropyl methyl sulfide smoothly reacted with the benzamides but gave the products in a bit lower yields than those of thioanisole (**5ba-5ma**).

Mechanistic studies. The deprotonation of thioanisole with LDA is the key step in alkylation of the benzamide, and it aroused our greatest concern. The direct deprotonation of thioanisole with LDA in THF, as expected, did not take place. Thus, benzamide should play an essential role in the deprotonation process. The nucleophilic acyl substitution reaction of benzamide with alkyl lithium should generate ketone intermediate and a new LDA. The α -H of the ketone would easily undergo the deprotonation reaction with LDA for its acidity. This easy enolization could provide the driving force for the entire reaction. To demonstrate the enolate formation, we carried out the reaction and quenched it with benzyl bromide rather than water. As expected, the further α -alkylation product **6a** was obtained in a good yield of 79% (Fig 2, eq 1).

After getting this thermodynamics validity, we attempted to find dynamics probability about the deprotonation of thioanisole with LDA. We compared the reaction of benzamide with thioanisole and thioanisole- d_3 precisely under exactly the same conditions at 0 °C. We monitored the reactions, measured the initiating reaction rates and found out a primary kinetic isotope effects (KIE = 3.2, eq 2-3), suggesting that the cleavage of the methyl C-H bond of thioanisole could be the rate determining step. The intramolecular competitive reaction between methyl C-H bond and C-D bond revealed a similar KIE value (KIE = 3.5, eq 4). More importantly, we found the reaction with thioanisole- d_3 gave the product with 60%

refer to SI

display

was there any carbon trapping exp?

refer SI

?

deuterium incorporation on the *ortho*-position of the carbonyl group (**3aa-d**), which revealed that the *ortho*-C–H bond of benzamide was involved in the reaction. To find out when the *ortho*-deuteration took place, we carried out the reaction of the ketone product **3aa** and thioanisole-*d*₃ and found no any deuterium scrambling product (eq 5), which excluded the deuteration of the ketone product. Hence, it is very possible that the *ortho*-deprotonation of the benzamide took place firstly to generate an aryl lithium intermediate, which underwent the deuteration reaction with thioanisole-*d*₃ before the reaction of benzamide.

Fig. 3 A Plausible Reaction Pathway. For clarity, intermediate A was shown in its monomer rather than a possible dimer structure.

On the bases of the results described above, we proposed a possible pathway for the substitution of benzamide **1a** with thioanisole-*d*₃ as shown in Fig 3. The deprotonation equilibrium between benzamide ($pK_a = 37.8$ in THF) and LDA (HDA: $pK_a = 35.7$ in THF)⁸⁴ provides *ortho*-lithiation intermediate **A**⁸⁵⁻⁸⁷ in a low concentration, which would undergo the σ -bond metathesis reaction with toluene-*d*₈ via a four-membered ring transition state (TS). The σ -bond metathesis reaction generates an alkyl lithium intermediate coordinating with the deuterated benzamide (**B**), which would easily undergo the nucleophilic acyl substitution reaction to give the ketone intermediate (**C**) and LDA. The easy acid-base reaction between the ketone and LDA could afford an enolate intermediate **D**, providing the driving force of the reaction. After quenching with water, the protonation of the enolate **D** finally afforded the product. The protonation of intermediate **B** with HDA could also give deuterated benzamide and deuterium-diluted toluene, resulting in the product having deuterium-diluted *ortho*-C–H bond.

Discussion

In summary, we have developed a direct alkylation reaction of unactivated benzamides with methyl sulfides, which provides an efficient strategy for the synthesis of α -sulfenylated ketones. This reaction features methyl C–H bond deprotonation of methyl sulfides with a relatively weak base LDA, which was actually not basic enough to deprotonate either methyl sulfides or benzamide. Preliminary mechanism studies revealed that the *ortho*-lithiation of benzamide promoted the deprotonation of methyl sulfides and triggered the nucleophilic acyl substitution reaction of the benzamide. Distinct from the amide activation strategies by either enhancing the amide reactivity or stabilizing the tetrahedral intermediates, the low concentration of the reactive carbanion intermediates arising from the deprotonation equilibrium of the weak base LDA could be the key factor for restraining side reactions and improving the selectivity. Further applications based on this unique process are currently ongoing in our group.

References:

- Greenberg, A., Breneman, C. M. & Liebman, J. F. *The amide linkage: selected structural aspects in chemistry, biochemistry, and materials science*; Wiley-Interscience: New York, 2000.
- Jin, Z. Amaryllidaceae and Sceletium alkaloids. *Nat. Prod. Rep.* **26**, 363-381 (2009).
- Froidevaux, V., Negrell, C., Caillol, S., Pascault, J. & Boutevin, B. Biobased Amines: From Synthesis to Polymers; Present and Future. *Chem. Rev.* **116**, 14181-14224 (2016).
- Brunton, L. L., Knollmann, B. C. & Hilal-Dandan, R. *Goodman & Gilman's: the pharmacological basis of therapeutics*; 3th ed., 2018.
- Braude, E. A. & Evans, E. A. Alkenylation with lithium alkenyls. Part XI. A new synthesis of ethylenic aldehydes. *J. Chem. Soc. (Resumed)*, 3334-3337 (1955).
- Olah, G. A., Surya Prakash, G. K. & Arvanaghi, M. *Synthetic Methods and Reactions*; Part 109. Improved Preparation of Aldehydes and Ketones from *N,N*-Dimethylamides and Grignard Reagents. *Synthesis*, 228-230 (1984).
- Pace, V., Holzer, W. & Olofsson, B. Increasing the Reactivity of Amides towards Organometallic Reagents: An Overview. *Adv. Synth. Catal.* **356**, 3697-3736 (2014).
- Ouyang, K., Hao, W., Zhang, W. & Xi, Z. Transition-Metal-Catalyzed Cleavage of C–N Single Bonds.

Supporting Information

Direct Alkylation of Unactivated Benzamides with Methyl Sulfides by Breaking Both C-N and C-H Bonds under

Transition-Metal-Free Condition

Can-Can Bao¹, Hui-Zhen Du¹, Yan-Long Luo¹ and Bing-Tao Guan^{2*}

¹College of Chemistry, Nankai University, Tianjin 300071, China

²Department of chemistry, Fudan University, 2005 Songhu Road, Shanghai 200438,
China

E-mail: bguan@fudan.edu.cn

Table of Contents

1. General Information	S2
2. Additional Condition Optimization	S3
3. General Procedures and Analytical Data of Compound 3 and 5	S4
4. Control Experiments	S26
5. Reference	S31
6. NMR Spectra	S32

1. General Information

All manipulations of air- and moisture-sensitive compounds were performed under a nitrogen atmosphere by use of standard Schlenk techniques or in a glovebox. THF, Et₂O, *t*-BuOMe, CPME and hexane, were dried by distillation over sodium/benzophenone. Lithium diisopropylamide (LDA) was synthesized according to literature¹, and kept as solid under -30 °C in a glove box after removing the solvents in vacuum. LDA purchased from MERYER Co., Ltd. was also used directly as received. Sulfides **2a**, **2e**, **2g**, **4a**, **4b**, **4c**, **4e**, **4f**, **4g** and **4h** purchased from Alfa Aesar, TCI (Shanghai) Development Co., Ltd., Shanghai Aladdin Biochemical Technology Co., Ltd. and Bidepharm were used as received. Sulfides **2b**, **2c**, **2h**, **2i** and **2l** were prepared from aryl bromides according to the literature.² Sulfides **2d**, **2f**, **2j**, **2k**, **2m** and **2n** were synthesized from thiophenols or phenols according to the literature.³ Sulfides **4d**, **4i**, **4j**, **4k**, **4l**, **4m**, **4n**, **4o** and **4p** were prepared from alkyl halides or their analogues according to the literature.⁴ Benzamides **1a**, **1b**, **1e**, **1f**, **1h**, **1j**, **1k**, **1l** and **1m** were synthesized from aryl chlorides according to the literature.⁵ Benzamides **1c**, **1d**, **1g** and **1i** were prepared from carboxylic acids according to the literature.⁶

TLC were performed on silica gel Huanghai HSGF254 plates and visualized by quenching of UV fluorescence (λ_{max} = 254 nm). Silica gel (200–300 mesh) was purchased from Qingdao Haiyang Chemical Co., China. ¹H NMR, ¹³C NMR were recorded on a Bruker ASCEND400 (400 MHz for ¹H, 101 MHz for ¹³C) instrument in CDCl₃ with tetramethylsilane as an internal standard. Chemical shifts (δ) are recorded in ppm relative to the residual solvent signals (CDCl₃: δ = 7.26 for ¹H, δ = 77.16 for ¹³C). Data were reported as follows: chemical shift in ppm (δ), multiplicity (s = singlet, d = doublet, t = triplet, q = quartet, m = multiplet), coupling constant (Hz), integration. Gas chromatography (GC) data were collected from SHIMADZU GAS CHROMATOGRAPH GC-2014 AOC-20i. High resolution mass spectra (HRMS) were recorded on a Varian 7.0T FTMS with Varian QFT-ESI or Agilent GCQTOF 7200 (EI).

MP measurement?

2. Additional Condition Optimization

Table S1. Alkylation of benzamide with different amount of LDA^a

which one? see S-2

$$\begin{array}{c}
 \text{Ph} \\
 | \\
 \text{N}(\text{Pr})_2 \\
 | \\
 \text{C}=\text{O} \\
 | \\
 \text{CH}_2 \\
 | \\
 \text{CH}_2 \\
 | \\
 \text{SPr} \\
 \text{2a}
 \end{array}
 \xrightarrow[\text{THF, 60 }^\circ\text{C, 12 h}]{\text{LDA (x equiv.)}}
 \begin{array}{c}
 \text{Ph} \\
 | \\
 \text{N}(\text{Pr})_2 \\
 | \\
 \text{C}=\text{O} \\
 | \\
 \text{CH}(\text{CH}_2\text{SPr}) \\
 | \\
 \text{CH}_2 \\
 \text{3aa}
 \end{array}$$

entry	LDA (x equiv.)	3aa yield (%) ^b
1	1.2	94
2	1.1	94
3	1.0	91
4	0.8	70
5	0.6	52
6	0.4	36
7 ^c	0.2	16

^aReaction Conditions: benzamide **1a** (0.50 mmol), thioanisole **2a** (1.00 mmol, 2.0 equiv.), LDA, THF (1.0 mL), 60 °C, 12 h. LDA: lithium diisopropylamide. ^bNMR yields and conversions with 2-methylxynaphthalene as an internal standard.

Table S2. Alkylation of benzamide with isopropyl methyl sulfide^a

$$\begin{array}{c}
 \text{Ph} \\
 | \\
 \text{N}(\text{Pr})_2 \\
 | \\
 \text{C}=\text{O} \\
 | \\
 \text{CH}_2 \\
 | \\
 \text{CH}_2 \\
 | \\
 \text{SPr} \\
 \text{4a}
 \end{array}
 \xrightarrow[\text{THF, 40 }^\circ\text{C, 24 h}]{\text{base (1.5 equiv.)}}
 \begin{array}{c}
 \text{Ph} \\
 | \\
 \text{N}(\text{Pr})_2 \\
 | \\
 \text{C}=\text{O} \\
 | \\
 \text{CH}(\text{CH}_2\text{SPr}) \\
 | \\
 \text{CH}_2 \\
 \text{5aa}
 \end{array}$$

entry	base	1a conv. (%) ^b	5aa yield (%) ^b
1	LiHMDS	< 5	0
2	KHMDS	< 5	0
3	LiTMP	> 95	80
4	LDA	82	82
5	n -BuLi ^c	> 95	16 ^d
6	LiCH ₂ TMS	> 95	24
7 ^e	LDA	93	92 (85)

^aReaction Conditions: benzamide **1a** (0.50 mmol), thioanisole **4a** (0.70 mmol, 1.4 equiv.), base (0.75 mmol, 1.5 equiv.), THF (1.0 mL), 40 °C, 24 h. HMDS: bis(trimethylsilyl)amide; LDA: lithium diisopropylamide; LiTMP: lithium 2,2,6,6-tetramethylpiperidine. ^bNMR yields and conversions with 1,3,5-trimethoxybenzene as an internal standard, isolated yield in parenthesis. ^c1.6 M in hexane. ^dButyl phenyl ketone was obtained as by-product (32% yield). ^eLDA (0.75 mmol), **4a** (1.25 mmol).

How about a tertiary alcohol side-product derived from the? ketones

S3

3. General Procedures and Analytical Data of Compound 3 and 5

3.1 A general procedure for the alkylation of benzamide with PhSMe

To a 25 mL Schlenk tube equipped with a Teflon septum and magnetic stir bar were added benzamide **1a** (102.7 mg, 0.50 mmol), methyl phenyl sulfide **2a** (82 μL, 0.70 mmol, 1.4 equiv.), THF (1.0 mL) and LDA (58.9 mg, 0.55 mmol, 1.1 equiv.). The tube was sealed and stirred at 40 °C for 24 hours. The mixture was then cooled to room temperature, quenched by adding five drops of H₂O and then diluted by ethyl acetate (EtOAc, 30 mL). The aqueous solution was dried over anhydrous Na₂SO₄. After filtration and concentration by rotary evaporation, the residue was purified by silica gel column chromatography (PE/EtOAc = 50/1) to afford the desired product **3aa** (106.2 mg, 0.46 mmol, 93% yield) as a colorless oil.

3.2 A general procedure for the alkylation of benzamide with *i*-PrSMe

To a 25 mL Schlenk tube equipped with a Teflon septum and magnetic stir bar were added benzamide **1a** (102.7 mg, 0.50 mmol), isopropyl methyl sulfide **4a** (136 μL, 1.25 mmol, 2.5 equiv.), THF (1.0 mL) and LDA (80.3 mg, 0.75 mmol, 1.5 equiv.). The tube was sealed and stirred at 40 °C for 24 hours. The mixture was then cooled to room temperature, quenched by adding five drops of H₂O and then diluted by ethyl acetate (EtOAc, 30 mL). The aqueous solution was dried over anhydrous Na₂SO₄. After filtration and concentration by rotary evaporation, the residue was purified by silica gel column chromatography (PE/EtOAc = 50/1) to afford the desired product **5aa** (82.6 mg, 0.43 mmol, 85% yield) as a colorless oil.

3.3 A gram-scale process for the alkylation of benzamide with thioanisole

To a 50 mL Schlenk tube equipped with a Teflon septum and magnetic stir bar were added benzamide **1a** (1.03 g, 5.0 mmol), methyl phenyl sulfide **2a** (825 μL, 7.0 mmol, 1.4 equiv.), THF (10 mL) and LDA (589.2 mg, 5.5 mmol, 1.1 equiv.). The tube was

S4

sealed and stirred at 40 °C for 24 hours. The mixture was then cooled to room temperature, quenched by saturated NH₄Cl aqueous solution (10 mL) and then diluted by ethyl acetate (EtOAc, 50 mL). The resulting mixture was extracted thrice with EtOAc (50 mL x 3). The combined organic layer was washed with brine (50 mL), and dried over anhydrous Na₂SO₄. After filtration and concentration by rotary evaporation, the mixture was purified by silica gel column chromatography (PE/EtOAc = 50/1) to afford the desired product **3aa** (1.06 g, 4.6 mmol, 92% yield) as a colorless oil.

3.4 Benzamides and sulfides that failed to give the desired products

3.5 Analytical Data of Compound 3 and 5

1-phenyl-2-((phenylthio)ethan-1-one (3aa).⁷ Colorless oil, 106.4 mg, 93% yield; ¹H NMR (400 MHz, CDCl₃) δ 7.94-7.92 (m, 2H), 7.58-7.54 (m, 1H), 7.46-7.42 (m, 2H), 7.38-7.36 (m, 2H), 7.28-7.24 (m, 2H), 7.22-7.18 (m, 1H), 4.26 (s, 2H); ¹³C NMR (101 MHz, CDCl₃) δ 194.1, 135.4, 134.8, 133.6, 130.5, 129.1, 128.74, 128.73, 127.1, 41.2; IR (liquid film, cm⁻¹) 3060, 3004, 2926, 1679, 1597, 1580, 1480, 1449, 1439, 1276, 1199, 1068, 975, 745, 688.

2-((4-ethylthio)phenyl)thio)-1-phenylethan-1-one (3ab). Pale pink oil, 138.8 mg, 96%

yield; ¹H NMR (400 MHz, CDCl₃) δ 7.95-7.93 (m, 2H), 7.58 (t, *J* = 7.2 Hz, 1H), 7.48-7.44 (m, 2H), 7.30 (d, *J* = 8.4 Hz, 2H), 7.21 (d, *J* = 8.4 Hz, 2H), 4.24 (s, 2H), 2.92 (q, *J* = 7.2 Hz, 2H), 1.30 (t, *J* = 7.2 Hz, 3H); ¹³C NMR (101 MHz, CDCl₃) δ 194.1, 136.4, 135.5, 133.6, 131.8, 131.6, 129.3, 128.8, 41.6, 27.6, 14.4; IR (liquid film, cm⁻¹) 3061, 2972, 2929, 2892, 2872, 1683, 1676, 1596, 1580, 1479, 1447, 1393, 1319, 1290, 1266, 1198, 1109, 998, 802, 749, 684, 640; HRMS (EI): *m/z* calcd. for C₁₆H₁₆O₂ [M]⁺: 288.0637, found: 288.0637.

2-((4-(dimethylamino)phenyl)thio)-1-phenylethan-1-one (3ac). Pale yellow oil, 122.9 mg, 91% yield; ¹H NMR (400 MHz, CDCl₃) δ 7.94-7.91 (m, 2H), 7.56 (t, *J* = 7.2 Hz, 1H), 7.47-7.43 (m, 2H), 7.30 (d, *J* = 8.4 Hz, 2H), 6.62 (d, *J* = 8.0 Hz, 2H), 4.07 (s, 2H), 2.95 (s, 6H); ¹³C NMR (101 MHz, CDCl₃) δ 194.7, 135.7, 135.3, 133.3, 128.9, 128.7, 112.9, 43.6, 40.5; IR (liquid film, cm⁻¹) 3061, 2894, 2810, 1676, 1595, 1506, 1447, 1359, 1276, 1226, 1195, 1130, 1100, 1014, 946, 813, 723, 689; HRMS (EI): *m/z* calcd. for C₁₆H₁₇NOS [M]⁺: 271.1025, found: 271.1021.

2-((4-isopropoxyphenyl)thio)-1-phenylethan-1-one (3ad). Colorless oil, 125.2 mg, 86% yield; ¹H NMR (400 MHz, CDCl₃) δ 7.93-7.90 (m, 2H), 7.59-7.55 (m, 1H), 7.47-7.43 (m, 2H), 7.35-7.32 (m, 2H), 6.81-6.78 (m, 2H), 4.56-4.47 (m, 1H), 4.14 (s, 2H), 1.32 (d, *J* = 5.6 Hz, 6H); ¹³C NMR (101 MHz, CDCl₃) δ 194.6, 158.3, 135.7, 134.8, 133.5, 128.9, 128.8, 124.3, 116.6, 70.1, 43.0, 22.1; IR (liquid film, cm⁻¹) 3063, 2977, 2933, 1679, 1592, 1490, 1449, 1385, 1281, 1245, 1182, 1117, 1103, 1016, 1001, 952, 828, 688; HRMS (ESI): *m/z* calcd. for C₁₇H₁₈NaO₂S [M + Na]⁺: 309.0920, found: 309.0925.

2-((4-methoxyphenyl)thio)-1-phenylethan-1-one (3ae).⁷ Colorless oil, 118.5 mg, 92% yield; ¹H NMR (400 MHz, CDCl₃) δ 7.92 (d, *J* = 7.6 Hz, 2H), 7.57 (t, *J* = 7.6 Hz, 1H), 7.48-7.44 (m, 2H), 7.36 (d, *J* = 8.8 Hz, 2H), 6.82 (d, *J* = 8.4 Hz, 2H), 4.13 (s, 2H), 3.78 (s, 3H); ¹³C NMR (101 MHz, CDCl₃) δ 194.5, 159.9, 135.6, 134.8, 133.5, 128.9, 128.8, 124.7, 114.8, 55.5, 42.9; IR (liquid film, cm⁻¹) 3064, 3004, 2959, 2939, 2837, 1678, 1592, 1493, 1462, 1449, 1288, 1248, 1173, 1104, 1029, 975, 827, 688.

2-((2-methoxyphenyl)thio)-1-phenylethan-1-one (3af).⁸ Colorless oil, 105.4 mg, 82% yield; ¹H NMR (400 MHz, CDCl₃) δ 7.85 (d, *J* = 7.6 Hz, 2H), 7.49-7.45 (m, 1H), 7.37-7.33 (m, 2H), 7.26 (d, *J* = 7.6 Hz, 1H), 7.18-7.14 (m, 1H), 6.80-6.75 (m, 2H), 4.14 (s, 2H), 3.75 (s, 3H); ¹³C NMR (101 MHz, CDCl₃) δ 194.6, 158.4, 135.7, 133.4, 132.7, 129.2, 128.71, 128.66, 122.1, 121.1, 110.8, 55.8, 39.6; IR (liquid film, cm⁻¹) 3060, 2964, 2938, 2837, 1687, 1597, 1579, 1474, 1464, 1450, 1271, 1239, 1096, 1059, 1021, 803, 741, 713, 688.

2-((3-methoxyphenyl)thio)-1-phenylethan-1-one (3ag).⁸ Colorless oil, 123.0 mg, 93% yield; ¹H NMR (400 MHz, CDCl₃) δ 7.96-7.94 (m, 2H), 7.60-7.56 (m, 1H), 7.48-7.45 (m, 2H), 7.21-7.17 (m, 1H), 6.97-6.93 (m, 2H), 6.77-6.75 (m, 1H), 4.29 (s, 2H), 3.77 (s, 3H); ¹³C NMR (101 MHz, CDCl₃) δ 194.2, 159.9, 136.2, 135.5, 133.6, 130.0, 128.81, 128.79, 122.4, 115.5, 113.0, 55.4, 41.1; IR (liquid film, cm⁻¹) 3063, 3004, 2937, 2835, 1682, 1590, 1576, 1479, 1449, 1283, 1248, 1232, 1039, 974, 860, 804, 777, 687.

2-((1,1'-biphenyl)-4-ylthio)-1-phenylethan-1-one (3ah).⁷ Colorless oil, 131.3 mg, 86% yield, m.p. 104-107 °C; ¹H NMR (400 MHz, CDCl₃) δ 7.99-7.97 (m, 2H), 7.62-7.42 (m, 11H), 7.37-7.34 (m, 1H), 4.32 (s, 2H); ¹³C NMR (101 MHz, CDCl₃) δ 194.2, 140.4, 140.2, 135.5, 133.9, 133.6, 131.0, 128.9, 128.8, 127.8, 127.6, 127.1, 41.3; IR (KBr, cm⁻¹) 3070, 3055, 2993, 2979, 1671, 1596, 1579, 1478, 1446, 1277, 1137, 1090, 1013, 1003, 938, 826, 764, 758, 737, 724, 701, 686, 647, 554; HRMS (ESI): *m/z* calcd. for C₂₀H₁₆NaOS [M + Na]⁺: 327.0814, found: 327.0819.

2-((1,1'-biphenyl)-3-ylthio)-1-phenylethan-1-one (3ai). Colorless oil, 128.4 mg, 83% yield; ¹H NMR (400 MHz, CDCl₃) δ 7.98-7.95 (m, 2H), 7.62-7.53 (m, 4H), 7.49-7.42 (m, 5H), 7.37-7.34 (m, 3H), 4.34 (s, 2H); ¹³C NMR (101 MHz, CDCl₃) δ 194.2, 142.2, 140.4, 135.5, 135.4, 133.6, 129.5, 129.22, 129.15, 128.9, 128.8, 127.7, 127.2, 126.1, 41.2; IR (liquid film, cm⁻¹) 3059, 3030, 2916, 2849, 1679, 1596, 1589, 1564, 1467, 1449, 1399, 1318, 1276, 1198, 1182, 1076, 1001, 991, 975, 756, 728, 698, 645, 614; HRMS (ESI): *m/z* calcd. for C₂₀H₁₆NaOS [M + Na]⁺: 327.0814, found: 327.0820.

2-((naphthalen-2-ylthio)-1-phenylethan-1-one (3aj).⁷ Colorless oil, 126.4 mg, 92% yield; ¹H NMR (400 MHz, CDCl₃) δ 7.98-7.96 (m, 2H), 7.84 (s, 1H), 7.80-7.73 (m, 3H), 7.61-7.57 (m, 1H), 7.49-7.43 (m, 5H), 4.38 (s, 2H); ¹³C NMR (101 MHz, CDCl₃) δ 194.2, 135.5, 133.8, 133.7, 132.32, 132.29, 129.0, 128.84, 128.80, 128.1, 127.8, 127.5, 126.7, 126.3, 41.3; IR (liquid film, cm⁻¹) 3081, 3052, 2943, 2902, 1687, 1586, 1501, 1447, 1372, 1337, 1287, 1195, 1076, 991, 859, 815, 783, 767, 746, 738, 688, 648, 554.

2-(naphthalen-1-ylthio)-1-phenylethan-1-one (3ak). Colorless oil, 130.9 mg, 93% yield; $^1\text{H NMR}$ (400 MHz, CDCl_3) δ 8.43 (d, $J = 8.4$ Hz, 2H), 7.91-7.85 (m, 3H), 7.79 (d, $J = 8.4$ Hz, 1H), 7.65 (dd, $J = 7.2, 0.8$ Hz, 1H), 7.58-5.0 (m, 3H), 7.44-7.36 (m, 3H), 4.30 (s, 2H); $^{13}\text{C NMR}$ (101 MHz, CDCl_3) δ 194.3, 135.6, 134.1, 133.5, 133.4, 131.7, 131.3, 128.9, 128.82, 128.78, 128.7, 126.9, 126.5, 125.7, 125.2, 41.8; IR (liquid film, cm^{-1}) 3054, 2963, 2927, 2893, 1679, 1591, 1561, 1448, 1383, 1317, 1289, 977, 783, 769, 749, 682, 665; HRMS (ESI): m/z calcd. for $\text{C}_{18}\text{H}_{14}\text{NaOS}$ [$\text{M} + \text{Na}^+$]: 301.0658, found: 301.0662.

(E)-1-phenyl-2-((4-styrylphenyl)thio)ethan-1-one (3al). White solid, 150.5 mg, 93% yield, m.p. 118-120 °C; $^1\text{H NMR}$ (400 MHz, CDCl_3) δ 7.96-7.94 (m, 2H), 7.60-7.57 (m, 1H), 7.50-7.41 (m, 6H), 7.37-7.33 (m, 4H), 7.28-7.24 (m, 1H), 7.10-7.01 (m, 2H), 4.28 (s, 2H); $^{13}\text{C NMR}$ (101 MHz, CDCl_3) δ 194.1, 137.2, 136.4, 135.5, 134.0, 133.7, 130.8, 129.2, 128.8, 127.9, 127.2, 126.7, 41.3; IR (KBr, cm^{-1}) 3073, 3052, 3019, 2923, 2880, 2852, 1675, 1596, 1588, 1580, 1495, 1447, 1394, 1280, 1186, 1090, 970, 807, 726, 710, 688, 530; HRMS (ESI): m/z calcd. for $\text{C}_{22}\text{H}_{18}\text{NaOS}$ [$\text{M} + \text{Na}^+$]: 353.0971, found: 353.0975.

2-((4-isopropylphenyl)thio)-1-phenylethan-1-one (3am). Colorless oil, 127.5 mg, 93% yield; $^1\text{H NMR}$ (400 MHz, CDCl_3) δ 7.95-7.93 (m, 2H), 7.59-7.55 (m, 1H), 7.47-7.43 (m, 2H), 7.35-7.33 (m, 2H), 7.16-7.14 (m, 2H), 4.24 (s, 2H), 2.93-2.82 (m, 1H), 1.23 (d, $J = 6.8$ Hz, 2H); $^{13}\text{C NMR}$ (101 MHz, CDCl_3) δ 194.4, 148.5, 135.6, 133.5, 131.5,

131.4, 128.8, 128.7, 127.4, 41.9, 33.8, 24.0; IR (liquid film, cm^{-1}) 3063, 2961, 2928, 2870, 1682, 1597, 1493, 1449, 1407, 1275, 1101, 1070, 1073, 1015, 1001, 974, 825, 715, 687; HRMS (ESI): m/z calcd. for $\text{C}_{17}\text{H}_{18}\text{NaOS}$ [$\text{M} + \text{Na}^+$]: 293.0971, found: 293.0975.

2-((4-(methylthio)phenyl)thio)-1-phenylethan-1-one (3an). White solid, 89.5 mg, 65% yield, m.p. 68.9-69.6 °C; $^1\text{H NMR}$ (400 MHz, CDCl_3) δ 7.94-7.92 (m, 2H), 7.58 (t, $J = 7.2$ Hz, 1H), 7.48-7.44 (m, 2H), 7.31 (d, $J = 8.0$ Hz, 2H), 7.15 (d, $J = 8.4$ Hz, 2H), 4.21 (s, 2H), 2.44 (s, 3H); $^{13}\text{C NMR}$ (101 MHz, CDCl_3) δ 194.1, 138.4, 135.4, 133.6, 131.9, 130.7, 128.8, 127.0, 41.8, 15.8; IR (KBr, cm^{-1}) 3326, 3067, 2995, 2951, 2918, 1677, 1596, 1577, 1494, 1430, 1390, 1278, 1185, 1107, 1016, 1008, 967, 802, 747, 727, 686, 655, 560; HRMS (ESI): m/z calcd. for $\text{C}_{15}\text{H}_{14}\text{NaOS}_2$ [$\text{M} + \text{Na}^+$]: 297.0378, found: 297.0382.

2,2'-(1,4-phenylenebis(sulfanediy))bis(1-phenylethan-1-one) (3ana). White solid, 105.7 mg, 93% yield, m.p. 125.2-126.3 °C; $^1\text{H NMR}$ (400 MHz, CDCl_3) δ 7.94-7.92 (m, 4H), 7.58 (t, $J = 7.2$ Hz, 2H), 7.49-7.45 (m, 4H), 7.29-7.28 (m, 4H), 4.26 (m, 4H); $^{13}\text{C NMR}$ (101 MHz, CDCl_3) δ 194.0, 135.4, 134.0, 133.7, 131.0, 128.9, 128.8, 41.2; IR (KBr, cm^{-1}) 3067, 3058, 2921, 2901, 2850, 1676, 1596, 1579, 1481, 1447, 1389, 1321, 1279, 1192, 1110, 1009, 998, 801, 753, 685, 640, 557; HRMS (ESI): m/z calcd. for $\text{C}_{22}\text{H}_{18}\text{NaO}_2\text{S}_2$ [$\text{M} + \text{Na}^+$]: 401.0640, found: 401.0642.

1-(4-(tert-butyl)phenyl)-2-(phenylthio)ethan-1-one (3ba). Colorless oil, 132.3 mg,

95% yield; ¹H NMR (400 MHz, CDCl₃) δ 7.90-7.88 (m, 2H), 7.48-7.46 (m, 2H), 7.40-7.37 (m, 2H), 7.29-7.25 (m, 2H), 7.23-7.18 (m, 1H), 4.26 (s, 2H), 1.34 (s, 9H); ¹³C NMR (101 MHz, CDCl₃) δ 193.8, 157.4, 135.1, 132.9, 130.4, 129.1, 128.8, 127.1, 125.8, 41.2, 35.3, 31.2; IR (liquid film, cm⁻¹) 3059, 2964, 2905, 2869, 1675, 1604, 1478, 1439, 1281, 1108, 1069, 999, 978, 745, 691; HRMS (ESI): *m/z* calcd. for C₁₈H₂₁OS [M + H]⁺: 285.1308, found: 285.1305.

1-(4-isopropylphenyl)-2-(phenylthio)ethan-1-one (3ca). Colorless oil, 127.3 mg, 99% yield; ¹H NMR (400 MHz, CDCl₃) δ 7.90-7.87 (m, 2H), 7.40-7.38 (m, 2H), 7.32-7.25 (m, 4H), 7.23-7.19 (m, 1H), 4.26 (s, 2H) (2.96 (m, 1H), 1.27 (d, *J* = 6.8 Hz, 6H); ¹³C NMR (101 MHz, CDCl₃) δ 193.8, 155.2, 135.1, 133.3, 130.4, 129.2, 129.1, 127.1, 126.9, 41.2, 34.4, 23.8; IR (liquid film, cm⁻¹) 3059, 2962, 2928, 1675, 1605, 1439, 1418, 1310, 1279, 1056, 1012, 999, 743, 691; HRMS (ESI): *m/z* calcd. for C₁₇H₁₉OS [M + H]⁺: 271.1151, found: 271.1151.

1-(4-cyclohexylphenyl)-2-(phenylthio)ethan-1-one (3da). White solid, 120.3 mg, 78% yield, m.p. 52.6-53.9 °C; ¹H NMR (400 MHz, CDCl₃) δ 7.88-7.86 (m, 2H), 7.39-7.37 (m, 2H), 7.29-7.24 (m, 4H), 7.22-7.18 (m, 1H), 4.25 (s, 2H), 2.58-2.53 (m, 1H), 1.89-1.81 (m, 4H), 1.78-1.74 (m, 1H), 1.47-1.33 (m, 4H), 1.30-1.20 (m, 1H); ¹³C NMR (101 MHz, CDCl₃) δ 193.8, 154.4, 135.1, 133.3, 130.4, 129.1, 129.0, 127.3, 127.1, 44.8, 41.2, 34.2, 26.8, 26.1; IR (KBr, cm⁻¹) 3058, 2925, 2851, 1686, 1603, 1584, 1481, 1451, 1369, 1293, 1213, 1189, 1089, 988, 820, 757, 731, 690, 567; HRMS (EI): *m/z* calcd. for C₂₀H₂₅OS [M]⁺: 310.1386, found: 310.1387.

1-(4-butylphenyl)-2-(phenylthio)ethan-1-one (3ea). Colorless oil, 115.4 mg, 74% yield; ¹H NMR (400 MHz, CDCl₃) δ 7.88-7.86 (m, 2H), 7.40-7.38 (m, 2H), 7.29-7.19 (m, 5H), 4.26 (s, 2H), 2.67 (t, *J* = 8.0 Hz, 2H), 1.65-1.58 (m, 2H), 1.40-1.31 (m, 2H), 0.93 (t, *J* = 7.6 Hz, 3H); ¹³C NMR (101 MHz, CDCl₃) δ 193.9, 149.5, 135.1, 133.2, 130.5, 129.2, 129.0, 128.9, 127.1, 41.3, 35.9, 33.3, 22.4, 14.0; IR (liquid film, cm⁻¹) 3059, 2956, 2930, 2859, 1675, 1605, 1466, 1439, 1416, 1280, 1179, 1069, 977, 744, 691; HRMS (ESI): *m/z* calcd. for C₁₈H₂₁OS [M + H]⁺: 285.1308, found: 285.1302.

1-(1,1'-biphenyl-4-yl)-2-(phenylthio)ethan-1-one (3fa) White solid, 136.2 mg, 89% yield, m.p. 89-92 °C; ¹H NMR (400 MHz, CDCl₃) δ 8.02-8.00 (m, 2H), 7.69-7.67 (m, 2H), 7.63-7.61 (m, 2H), 7.49-7.45 (m, 2H), 7.42-7.39 (m, 3H), 7.31-7.26 (m, 2H), 7.25-7.20 (m, 1H), 4.29 (s, 2H); ¹³C NMR (101 MHz, CDCl₃) δ 193.8, 146.2, 139.8, 134.9, 134.1, 130.6, 129.4, 129.2, 129.1, 128.5, 127.40, 127.38, 127.2, 41.3; IR (KBr, cm⁻¹) 3054, 2946, 2923, 2905, 1686, 1601, 1580, 1481, 1437, 1402, 1386, 1204, 1192, 1129, 981, 846, 764, 696, 689; HRMS (ESI): *m/z* calcd. for C₂₀H₁₉NaOS [M + Na]⁺: 327.0814, found: 327.0820.

1-(1,1'-biphenyl-3-yl)-2-(phenylthio)ethan-1-one (3ga). Colorless oil, 130.1 mg, 89% yield; ¹H NMR (400 MHz, CDCl₃) δ 8.15-8.14 (m, 1H), 7.93-7.90 (m, 1H), 7.81-7.78 (m, 1H), 7.59-7.57 (m, 2H), 7.55-7.51 (m, 1H), 7.48-7.44 (m, 2H), 7.42-7.36 (m, 3H), 7.30-7.20 (m, 3H), 4.31 (s, 2H); ¹³C NMR (101 MHz, CDCl₃) δ 194.1, 141.9, 140.0, 135.9, 134.8, 132.2, 130.7, 129.23, 129.19, 129.0, 128.0, 127.6, 127.5, 127.3, 41.4; IR

(liquid film, cm^{-1}) 3059, 3032, 2926, 1682, 1583, 1478, 1453, 1439, 1301, 1074, 1025, 983, 745, 697; HRMS (ESI): m/z calcd. for $\text{C}_{20}\text{H}_{16}\text{NaOS}$ [$\text{M} + \text{Na}^+$]: 327.0814, found: 327.0819.

1-(4-(dimethylamino)phenyl)-2-(phenylthio)ethan-1-one (3ha). Pale yellow oil, 116.0 mg, 85% yield; ^1H NMR (400 MHz, CDCl_3) δ 7.89-7.86 (m, 2H), 7.42-7.40 (m, 2H), 7.30-7.26 (m, 2H), 7.22-7.18 (m, 1H), 6.66-6.63 (m, 2H), 4.23 (s, 2H), 3.06 (s, 6H); ^{13}C NMR (101 MHz, CDCl_3) δ 192.2, 153.7, 135.9, 131.1, 130.0, 129.0, 126.7, 123.3, 110.7, 40.7, 40.1; IR (liquid film, cm^{-1}) 3056, 2908, 2817, 1655, 1597, 1549, 1530, 1481, 1438, 1374, 1322, 1293, 1231, 1191, 1122, 1024, 945, 820, 743, 691; HRMS (EI): m/z calcd. for $\text{C}_{16}\text{H}_{17}\text{NOS}$ [M^+]: 271.1025, found: 271.1024.

1-(3-phenoxyphenyl)-2-(phenylthio)ethan-1-one (3ia). Colorless oil, 160.2 mg, 82% yield; ^1H NMR (400 MHz, CDCl_3) δ 7.66-7.64 (m, 1H), 7.57-7.56 (m, 1H), 7.44-7.34 (m, 5H), 7.30-7.20 (m, 4H), 7.17-7.13 (m, 1H), 7.03-7.01 (m, 2H), 4.22 (s, 2H); ^{13}C NMR (101 MHz, CDCl_3) δ 193.6, 158.0, 156.6, 137.3, 134.7, 130.8, 130.2, 130.1, 129.2, 127.4, 124.1, 123.7, 123.5, 119.3, 118.6, 41.4; IR (liquid film, cm^{-1}) 3063, 2916, 1682, 1581, 1489, 1456, 1439, 1273, 1236, 1164, 1071, 1024, 912, 894, 745, 691; HRMS (ESI): m/z calcd. for $\text{C}_{20}\text{H}_{16}\text{NaOS}$ [$\text{M} + \text{Na}^+$]: 343.0763, found: 343.0768.

1-(4-methoxyphenyl)-2-(phenylthio)ethan-1-one (3ja).⁷ Colorless oil, 121.2 mg, 91% yield; ^1H NMR (400 MHz, CDCl_3) δ 7.94-7.92 (m, 2H), 7.40-7.37 (m, 2H), 7.29-7.26 (m, 2H), 7.23-7.19 (m, 1H), 6.94-6.91 (m, 2H), 4.23 (s, 2H), 3.86 (s, 3H); ^{13}C NMR

(101 MHz, CDCl_3) δ 192.8, 163.9, 135.2, 131.2, 130.4, 129.2, 128.5, 127.1, 114.0, 55.6, 41.1; IR (liquid film, cm^{-1}) 3052, 3015, 2978, 2934, 2905, 1659, 1603, 1574, 1509, 1421, 1313, 1265, 1202, 1181, 1024, 994, 830, 818, 741, 690.

1-(3-methoxyphenyl)-2-(phenylthio)ethan-1-one (3ka). Colorless oil, 115.5 mg, 90% yield; ^1H NMR (400 MHz, CDCl_3) δ 7.53-7.47 (m, 2H), 7.41-7.35 (m, 3H), 7.31-7.21 (m, 3H), 7.14-7.11 (m, 1H), 4.27 (s, 2H), 3.84 (s, 3H); ^{13}C NMR (101 MHz, CDCl_3) δ 194.0, 160.0, 136.9, 134.9, 130.7, 129.8, 129.2, 127.3, 121.4, 120.2, 113.0, 55.6, 41.4; IR (liquid film, cm^{-1}) 3074, 3060, 3004, 2960, 2940, 2836, 1679, 1597, 1582, 1487, 1465, 1438, 1279, 1222, 1069, 1024, 789, 744, 690; HRMS (ESI): m/z calcd. for $\text{C}_{15}\text{H}_{14}\text{NaO}_2\text{S}$ [$\text{M} + \text{Na}^+$]: 281.0607, found: 281.0611.

1-(3,4-dimethoxyphenyl)-2-(phenylthio)ethan-1-one (3la). Colorless oil, 136.3 mg, 92% yield; ^1H NMR (400 MHz, CDCl_3) δ 7.56 (dd, $J = 8.4, 2.0$ Hz, 1H), 7.50 (d, $J = 2.0$ Hz, 1H), 7.41-7.39 (m, 2H), 7.30-7.26 (m, 2H), 7.23-7.20 (m, 1H), 6.87 (d, $J = 8.4$ Hz, 1H), 4.24 (s, 2H), 3.94 (s, 3H), 3.90 (s, 3H); ^{13}C NMR (101 MHz, CDCl_3) δ 192.9, 153.7, 149.2, 135.2, 130.4, 129.1, 128.6, 127.1, 123.6, 110.8, 110.1, 56.2, 56.1, 40.8; IR (liquid film, cm^{-1}) 3078, 3059, 3004, 2961, 2936, 2839, 1668, 1594, 1585, 1515, 1464, 1296, 1273, 1151, 1022, 745, 692; HRMS (ESI): m/z calcd. for $\text{C}_{16}\text{H}_{16}\text{NaO}_3\text{S}$ [$\text{M} + \text{Na}^+$]: 311.0712, found: 311.0718.

1-(naphthalen-2-yl)-2-(phenylthio)ethan-1-one (3ma) colorless White solid, 113.7 mg, 83% yield, m.p. 69-72 °C; ^1H NMR (400 MHz, CDCl_3) δ 8.41 (s, 1H), 7.99 (dd, $J = 8.8, 2.0$

Hz, 1H), 7.91-7.84 (m, 3H), 7.61-7.57 (m, 1H), 7.55-7.51 (m, 1H), 7.42-7.40 (m, 2H), 7.29-7.19 (m, 3H), 4.38 (s, 2H); ^{13}C NMR (101 MHz, CDCl_3) δ 194.2, 135.8, 134.9, 132.8, 132.5, 130.73, 130.68, 129.7, 129.2, 128.8, 128.7, 127.9, 127.3, 127.0, 124.3, 41.4; IR (KBr, cm^{-1}) 3055, 2925, 1697, 1670, 1626, 1470, 1437, 1284, 1138, 1116, 1022, 957, 825, 781, 742, 689; HRMS (ESI): m/z calcd. for $\text{C}_{18}\text{H}_{14}\text{NaOS}$ [$\text{M} + \text{Na}$] $^+$: 301.0658, found: 301.0660.

2-(isopropylthio)-1-phenylethan-1-one (5aa).⁹ Colorless oil, 82.0 mg, 85% yield; ^1H NMR (400 MHz, CDCl_3) δ 7.98-7.96 (m, 2H), 7.58-7.54 (m, 1H), 7.48-7.44 (m, 2H), 3.83 (s, 2H), 3.03-2.93 (m, 1H), 1.27 (d, $J = 6.8$ Hz, 6H); ^{13}C NMR (101 MHz, CDCl_3) δ 195.1, 135.4, 133.4, 128.9, 128.7, 36.4, 35.6, 23.0; IR (liquid film, cm^{-1}) 3062, 2962, 2925, 2866, 1677, 1598, 1580, 1449, 1384, 1367, 1307, 1278, 1247, 1068, 1015, 971, 726, 688.

2-(tert-butylthio)-1-phenylethan-1-one (5ab). Colorless oil, 92.7 mg, 89% yield; ^1H NMR (400 MHz, CDCl_3) δ 7.95 (d, $J = 8.0$ Hz, 2H), 7.57-7.53 (m, 1H), 7.47-7.43 (m, 2H), 3.88 (s, 2H), 1.35 (s, 9H); ^{13}C NMR (101 MHz, CDCl_3) δ 196.4, 135.7, 133.4, 128.9, 128.7, 43.8, 35.8, 30.8; IR (liquid film, cm^{-1}) 3063, 2963, 2923, 2898, 1678, 1597, 1449, 1364, 1276, 1219, 1165, 1111, 1014, 1001, 971, 719, 688; HRMS (ESI): m/z calcd. for $\text{C}_{12}\text{H}_{16}\text{NaOS}$ [$\text{M} + \text{Na}$] $^+$: 231.0814, found: 231.0820.

2-(cyclohexylthio)-1-phenylethan-1-one (5ac). Colorless oil, 93.1 mg, 79% yield; ^1H NMR (400 MHz, CDCl_3) δ 7.97 (d, $J = 7.6$ Hz, 2H), 7.58-7.55 (m, 1H), 7.48-7.44 (m,

2H), 3.82 (s, 2H), 2.77-2.70 (m, 1H), 2.01-1.98 (m, 2H), 1.75-1.72 (m, 2H), 1.42-1.20 (m, 6H); ^{13}C NMR (101 MHz, CDCl_3) δ 195.3, 135.4, 133.4, 128.9, 128.7, 44.0, 35.8, 33.2, 26.0, 25.9; IR (liquid film, cm^{-1}) 3063, 2929, 2852, 1678, 1597, 1579, 1449, 1278, 1179, 1099, 1071, 1017, 999, 971, 887, 845, 817, 720, 687; HRMS (ESI): m/z calcd. for $\text{C}_{14}\text{H}_{18}\text{NaOS}$ [$\text{M} + \text{Na}$] $^+$: 257.0971, found: 257.0975.

2-(cycloheptylthio)-1-phenylethan-1-one (5ad). Colorless oil, 94.5 mg, 76% yield; ^1H NMR (400 MHz, CDCl_3) δ 7.98-7.96 (m, 2H), 7.56 (tt, $J = 7.2$, 1.6 Hz, 1H), 7.48-7.44 (m, 2H), 3.79 (s, 2H), 2.95-2.88 (m, 1H), 2.03-1.96 (m, 2H), 1.71-1.63 (m, 2H), 1.60-1.40 (m, 8H); ^{13}C NMR (101 MHz, CDCl_3) δ 195.1, 135.4, 133.3, 128.9, 128.7, 45.5, 36.6, 34.5, 28.4, 25.8; IR (liquid film, cm^{-1}) 3061, 2926, 2854, 1675, 1598, 1580, 1448, 1315, 1276, 1197, 1181, 1159, 1071, 1014, 972, 804, 726, 688; HRMS (EI): m/z calcd. for $\text{C}_{15}\text{H}_{20}\text{OS}$ [M] $^+$: 248.1229, found: 248.1228.

2-(methylthio)-1-phenylethan-1-one (5ae).¹⁰ Colorless oil, 58.2 mg, 70% yield; ^1H NMR (400 MHz, CDCl_3) δ 7.99-7.96 (m, 2H), 7.57 (tt, $J = 7.2$, 1.6 Hz, 1H), 7.49-7.45 (m, 2H), 3.76 (s, 2H), 2.14 (s, 3H); ^{13}C NMR (101 MHz, CDCl_3) δ 194.2, 135.3, 133.4, 128.9, 128.8, 39.2, 16.0; IR (liquid film, cm^{-1}) 3062, 2981, 2920, 1674, 1597, 1580, 1492, 1449, 1317, 1279, 1199, 1182, 1073, 1019, 1001, 979, 808, 729, 704, 687, 646.

1-phenyl-2-(propylthio)ethan-1-one (5af).⁷ Colorless oil, 65.4 mg, 67% yield; ^1H NMR (400 MHz, CDCl_3) δ 7.98-7.96 (m, 2H), 7.58-7.54 (m, 1H), 7.48-7.44 (m, 2H), 3.77 (s, 2H), 2.53 (t, $J = 7.3$ Hz, 2H), 1.66-1.57 (m, 2H), 0.96 (t, $J = 7.6$ Hz, 3H); ^{13}C

NMR (101 MHz, CDCl₃) δ 194.7, 135.3, 133.4, 128.9, 128.7, 37.1, 34.4, 22.4, 13.4; IR (liquid film, cm⁻¹) 3061, 2963, 2931, 2872, 1675, 1598, 1580, 1448, 1417, 1377, 1315, 1277, 1197, 1182, 1138, 1075, 1014, 974, 806, 728, 689, 647, 563.

2-(pentylthio)-1-phenylethanol-1-one (5ag). Colorless oil, 70.3 mg, 65% yield; ¹H NMR (400 MHz, CDCl₃) δ 7.99-7.97 (m, 2H), 7.59-7.55 (m, 3H), 7.49-7.45 (m, 2H), 3.78 (s, 2H), 2.56 (t, J = 7.2 Hz, 2H), 1.63-1.56 (m, 2H), 1.38-1.26 (m, 4H), 0.88 (t, J = 6.4 Hz); ¹³C NMR (101 MHz, CDCl₃) δ 194.7, 135.4, 133.4, 128.9, 128.8, 37.3, 32.5, 31.0, 28.8, 22.4, 14.1; IR (liquid film, cm⁻¹) 3061, 2956, 2928, 2871, 2858, 1675, 1598, 1580, 1466, 1448, 1417, 1315, 1277, 1197, 1182, 1137, 1075, 1015, 805, 749, 727, 689, 647, 563; HRMS (ESI): m/z calcd. for C₁₃H₁₈NaOS [M]⁺: 245.0975.

2-(dodecylthio)-1-phenylethanol-1-one (5ah).⁷ Colorless oil, 122.1 mg, 76% yield; ¹H NMR (400 MHz, CDCl₃) δ 7.97 (d, J = 7.6 Hz, 2H), 7.58-7.55 (m, 1H), 7.48-7.44 (m, 2H), 3.77 (s, 2H), 2.55 (t, J = 7.6 Hz, 2H), 1.62-1.54 (m, 2H), 1.36-1.24 (m, 18H), 0.88 (t, J = 6.8 Hz, 3H); ¹³C NMR (101 MHz, CDCl₃) δ 194.7, 135.4, 133.4, 128.9, 128.8, 37.2, 32.5, 32.0, 29.8, 29.7, 29.6, 29.5, 29.3, 29.1, 28.9, 22.8, 14.2; IR (KBr, cm⁻¹) 3061, 2954, 2923, 2852, 1673, 1597, 1580, 1490, 1467, 1450, 1378, 1275, 1176, 1096, 1076, 978, 832, 720, 685, 655.

2-(cyclopentylmethylthio)-1-phenylethanol-1-one (5ai). Colorless oil, 73.5 mg, 63% yield; ¹H NMR (400 MHz, CDCl₃) δ 7.99-7.96 (m, 2H), 7.59-7.55 (m, 1H), 7.48-7.45 (m, 2H), 3.78 (s, 2H), 2.57 (d, J = 7.6 Hz, 2H), 2.12-2.01 (m, 2H), 1.83-1.76 (m, 2H),

1.64-1.48 (m, 4H), 1.25-1.16 (m, 2H); ¹³C NMR (101 MHz, CDCl₃) δ 194.7, 135.4, 133.4, 128.9, 128.8, 39.4, 38.7, 37.7, 32.4, 25.3; IR (liquid film, cm⁻¹) 3062, 2924, 2851, 1675, 1598, 1580, 1492, 1448, 1415, 1316, 1278, 1182, 1070, 1014, 974, 928, 893, 806, 719, 688, 646; HRMS (EI): m/z calcd. for C₁₄H₁₈OS [M]⁺: 234.1073, found: 234.1070.

2-(cyclohexylmethylthio)-1-phenylethanol-1-one (5aj). Colorless oil, 87.7 mg, 71% yield; ¹H NMR (400 MHz, CDCl₃) δ 7.98-7.96 (m, 2H), 7.59-7.54 (m, 1H), 7.48-7.44 (m, 2H), 3.75 (s, 2H), 2.45 (d, J = 6.8 Hz, 2H), 1.82-1.79 (m, 2H), 1.71-1.61 (m, 3H), 1.53-1.42 (m, 1H), 1.26-1.06 (m, 3H), 0.96-0.86 (m, 2H); ¹³C NMR (101 MHz, CDCl₃) δ 194.7, 135.4, 133.4, 128.9, 128.7, 39.9, 37.7, 37.4, 32.8, 26.4, 26.1; IR (liquid film, cm⁻¹) 3062, 2924, 2851, 1675, 1598, 1580, 1492, 1448, 1415, 1316, 1278, 1182, 1070, 1014, 974, 928, 893, 806, 719, 688, 646; HRMS (EI): m/z calcd. for C₁₅H₂₀OS [M]⁺: 248.1229, found: 248.1223.

2-(((3R,5R)-adamantan-1-yl)methylthio)-1-phenylethanol-1-one (5ak). Colorless oil, 95.1 mg, 63% yield; ¹H NMR (400 MHz, CDCl₃) δ 7.98-7.96 (m, 2H), 7.59-7.54 (m, 1H), 7.48-7.44 (m, 2H), 3.75 (s, 2H), 2.40 (s, 2H), 1.95-1.94 (m, 3H), 1.63 (dd, J = 31.6, 12 Hz, 6H), 1.53 (d, J = 2.8 Hz, 6H); ¹³C NMR (101 MHz, CDCl₃) δ 194.8, 135.4, 133.3, 128.9, 128.7, 47.4, 41.8, 39.0, 36.9, 34.1, 28.6; IR (liquid film, cm⁻¹) 3061, 2902, 2846, 1675, 1598, 1580, 1449, 1360, 1344, 1316, 1276, 1196, 1181, 1099, 1015, 976, 725, 688; HRMS (EI): m/z calcd. for C₁₉H₂₄OS [M]⁺: 300.1542, found: 300.1546.

1-phenyl-2-((3-phenylpropyl)thio)ethanol-1-one (5al).¹¹ Colorless oil, 92.6 mg, 69%

yield; ¹H NMR (400 MHz, CDCl₃) δ 7.98-7.95 (m, 2H), 7.57 (t, *J* = 7.6 Hz, 1H), 7.48-7.44 (m, 2H), 7.28-7.24 (m, 2H), 7.19-7.14 (m, 3H), 3.78 (s, 2H), 2.68 (t, *J* = 7.6 Hz, 2H), 2.58 (t, *J* = 7.2 Hz, 2H), 1.91 (tt, *J* = 7.6, 7.2 Hz, 2H); ¹³C NMR (101 MHz, CDCl₃) δ 194.6, 141.4, 135.3, 133.4, 128.9, 128.8, 128.6, 126.0, 37.2, 34.8, 31.8, 30.6; IR (liquid film, cm⁻¹) 3061, 3026, 2927, 2855, 1675, 1598, 1580, 1496, 1449, 1416, 1277, 1198, 1181, 1075, 1015, 974, 746, 700, 688.

5am

1-phenyl-2-(3-(trimethylsilyl)propylthio)ethan-1-one (5am). Colorless oil, 94.5 mg, 70% yield; ¹H NMR (400 MHz, CDCl₃) δ 7.99-7.96 (m, 2H), 7.58-7.54 (m, 1H), 7.48-7.44 (m, 2H), 3.77 (s, 2H), 2.57 (t, *J* = 7.2 Hz, 2H), 1.62-1.54 (m, 2H), 0.58-0.54 (m, 2H), -0.03 (s, 9H); ¹³C NMR (101 MHz, CDCl₃) δ 194.7, 135.3, 133.4, 128.9, 128.81, 128.75, 37.1, 36.1, 23.9, 16.3, -1.6; IR (liquid film, cm⁻¹) 3063, 2952, 2921, 1676, 1598, 1580, 1492, 1449, 1415, 1306, 1277, 1248, 1415, 1306, 1277, 1248, 1197, 1181, 1016, 1001, 975, 863, 837, 727, 689, 646; HRMS (EI): *m/z* calcd. for C₁₄H₂₀OSSi [M]⁺: 266.1155, found: 266.1158.

5an

2-(4-methoxybutylthio)-1-phenylethan-1-one (5an). Colorless oil, 74.6 mg, 63% yield; ¹H NMR (400 MHz, CDCl₃) δ 7.98 (d, *J* = 7.2 Hz, 2H), 7.58 (t, *J* = 7.2 Hz, 1H), 7.49-7.45 (m, 2H), 3.79 (s, 2H), 3.36 (t, *J* = 6.0 Hz, 2H), 2.59 (t, *J* = 6.8 Hz, 2H), 1.70-1.63 (m, 4H); ¹³C NMR (101 MHz, CDCl₃) δ 194.6, 135.3, 133.4, 128.8, 128.7, 72.2, 58.6, 37.1, 32.1, 28.7, 25.7; IR (liquid film, cm⁻¹) 3060, 2930, 2866, 2827, 1675, 1630, 1598, 1580, 1448, 1418, 1388, 1315, 1277, 1198, 1117, 1014, 885, 806, 728, 689; HRMS (EI): *m/z* calcd. for C₁₃H₁₈O₂S [M]⁺: 238.1022, found: 238.1024.

5ao

2-(4-(tert-butylidimethylsilyloxy)butylthio)-1-phenylethan-1-one (5ao). Colorless oil, 105.7 mg, 62% yield; ¹H NMR (400 MHz, CDCl₃) δ 7.98-7.95 (m, 2H), 7.56 (tt, *J* = 7.2, 2.0 Hz, 1H), 7.48-7.44 (m, 2H), 3.77 (s, 2H), 3.59 (t, *J* = 6.4 Hz, 2H), 2.58 (t, *J* = 7.2 Hz, 2H), 1.69-1.54 (m, 4H), 0.87 (m, 9H), 0.03 (s, 6H); ¹³C NMR (101 MHz, CDCl₃) δ 194.6, 135.3, 133.4, 128.9, 128.7, 62.6, 37.1, 32.3, 31.9, 26.0, 25.4, 18.4, -5.2; IR (liquid film, cm⁻¹) 3060, 2953, 2929, 2857, 1676, 1598, 1581, 1472, 1449, 1388, 1277, 1256, 1102, 1014, 837, 776, 727, 688; HRMS (EI): *m/z* calcd. For C₁₈H₃₀O₂SSi [M]⁺: 338.1730, found: 338.1735.

5ap

2-(3-(dimethylamino)propylthio)-1-phenylethan-1-one (5ap). Colorless oil, 98.3 mg, 83% yield; ¹H NMR (400 MHz, CDCl₃) δ 7.96-7.94 (m, 2H), 7.56-7.53 (m, 1H), 7.46-7.42 (m, 2H), 3.78 (s, 2H), 3.58 (t, *J* = 7.2 Hz, 2H), 2.30 (t, *J* = 7.2 Hz, 2H), 2.18 (s, 6H), 1.74 (tt, *J* = 7.2, 7.2 Hz, 2H); ¹³C NMR (101 MHz, CDCl₃) δ 194.5, 135.3, 133.4, 128.8, 128.7, 58.4, 45.5, 37.2, 30.3, 27.1; IR (liquid film, cm⁻¹) 3060, 2940, 2858, 2816, 2767, 1675, 1598, 1580, 1448, 1377, 1277, 1199, 1042, 1013, 727, 689; HRMS (EI): *m/z* calcd. For C₁₃H₁₉NOS [M]⁺: 237.1182, found: 237.1185.

1-(4-(tert-butyl)phenyl)-2-(isopropylthio)ethan-1-one (5ba). Colorless oil, 100.9 mg, 81% yield; ¹H NMR (400 MHz, CDCl₃) δ 7.92 (d, *J* = 8.4 Hz, 2H), 7.48 (d, *J* = 8.8 Hz, 2H), 3.81 (s, 2H), 3.05-2.95 (m, 1H), 1.34 (s, 9H), 1.28 (d, *J* = 6.4 Hz, 6H); ¹³C NMR (101 MHz, CDCl₃) δ 194.8, 157.2, 132.8, 128.9, 125.7, 36.3, 35.6, 35.3, 31.2, 23.1; IR (liquid film, cm⁻¹) 3060, 2940, 2858, 2816, 2767, 1675, 1598, 1580, 1448, 1377, 1277,

1199, 1042, 1013, 727, 689; HRMS (EI): m/z calcd. For $C_{15}H_{22}OS$ [M]⁺: 250.1386, found: 250.1385.

~~1-(4-isopropylphenyl)-2-(isopropylthio)ethan-1-one (5ca)~~

~~1-(4-isopropylphenyl)-2-(isopropylthio)ethan-1-one (5ca)~~. Colorless oil, 86.6 mg, 73% yield; ¹H NMR (400 MHz, CDCl₃) δ 7.91 (d, J = 8.4 Hz, 2H), 7.31 (d, J = 8.0 Hz, 2H), 3.80 (s, 2H), 3.04-2.91 (m, 2H), 1.28-1.25 (m, 12H); ¹³C NMR (101 MHz, CDCl₃) δ 194.8, 154.9, 133.3, 129.1, 126.8, 36.3, 35.6, 34.4, 23.7, 23.0; IR (liquid film, cm⁻¹) 3049, 3030, 2962, 2928, 2869, 1674, 1606, 1569, 1463, 1419, 1384, 1365, 1310, 1279, 1186, 1055, 1011, 974, 851, 826, 781; HRMS (EI): m/z calcd. For $C_{14}H_{20}OS$ [M]⁺: 236.1229, found: 236.1229.

~~1-(4-cyclohexylphenyl)-2-(isopropylthio)ethan-1-one (5da)~~. Colorless oil, 108.4 mg, 79% yield; ¹H NMR (400 MHz, CDCl₃) δ 7.90 (d, J = 8.4 Hz, 2H), 7.29 (d, J = 8.0 Hz, 2H), 3.80 (s, 2H), 3.04-2.94 (m, 1H), 2.59-2.52 (m, 1H), 1.89-1.83 (m, 4H), 1.78-1.74 (m, 1H), 1.48-1.34 (m, 4H), 1.30-1.27 (m, 7H); ¹³C NMR (101 MHz, CDCl₃) δ 194.8, 154.1, 133.2, 129.1, 127.2, 44.8, 36.3, 35.6, 34.2, 26.8, 26.1, 23.1; IR (liquid film, cm⁻¹) 3030, 2926, 2852, 1672, 1605, 1569, 1449, 1419, 1383, 1366, 1310, 1279, 1247, 1185, 1155, 1056, 1026, 1011, 999, 844, 775, 714, 657; HRMS (EI): m/z calcd. For $C_{17}H_{24}OS$ [M]⁺: 276.1542, found: 276.1546.

1-(4-butylphenyl)-2-(isopropylthio)ethan-1-one (5ea). Colorless oil, 43.5 mg, 36% yield; ¹H NMR (400 MHz, CDCl₃) δ 7.91-7.89 (m, 2H), 7.28-7.26 (m, 2H), 3.81 (s,

2H), 3.05-2.95 (m, 1H), 2.67 (t, J = 8.0 Hz, 2H), 1.65-1.58 (m, 2H), 1.41-1.32 (m, 2H), 1.28 (d, J = 6.8 Hz, 6H), 0.93 (t, J = 7.4 Hz, 3H); ¹³C NMR (101 MHz, CDCl₃) δ 194.9, 149.2, 133.1, 129.0, 128.8, 36.3, 35.8, 35.6, 33.3, 23.1, 22.5, 14.0; IR (liquid film, cm⁻¹) 3030, 2926, 2852, 1672, 1605, 1569, 1449, 1419, 1383, 1366, 1311, 1279, 1185, 1155, 1056, 1026, 1011, 999, 844, 775, 714, 657; HRMS (EI): m/z calcd. For $C_{15}H_{22}OS$ [M]⁺: 250.1386, found: 250.1390.

~~1-(1,1'-biphenyl-4-yl)-2-(isopropylthio)ethan-1-one (5fa)~~. White solid, 107.2 mg, 80% yield, m.p. 66.8-67.3 °C; ¹H NMR (400 MHz, CDCl₃) δ 8.06 (d, J = 8.4 Hz, 2H), 7.70-7.68 (m, 2H), 7.64-7.62 (m, 2H), 7.50-7.46 (m, 2H), 7.43-7.39 (m, 1H), 3.86 (s, 2H), 3.07-2.97 (m, 1H), 1.31 (d, J = 6.8 Hz, 6H); ¹³C NMR (101 MHz, CDCl₃) δ 194.7, 146.0, 139.9, 134.1, 129.5, 129.1, 128.4, 127.37, 127.37, 36.4, 35.6, 23.1; IR (KBr, cm⁻¹) 3059, 2979, 2965, 2936, 2870, 1671, 1603, 1581, 1486, 1449, 1418, 1361, 1315, 1289, 1273, 1247, 1182, 1156, 1056, 1027, 1004, 856, 755, 692, 680, 569; HRMS (EI): m/z calcd. For $C_{17}H_{18}OS$ [M]⁺: 270.1073, found: 270.1075.

~~1-(1,1'-biphenyl-3-yl)-2-(isopropylthio)ethan-1-one (5ga)~~. Colorless oil, 125.7 mg, 93% yield; ¹H NMR (400 MHz, CDCl₃) δ 8.21 (s, 1H), 7.96-7.94 (m, 1H), 7.81-7.79 (m, 1H), 7.63-7.61 (m, 2H), 7.55 (t, J = 8.0 Hz, 1H), 7.49-7.46 (m, 2H), 7.41-7.37 (m, 1H), 3.88 (s, 2H), 3.07-2.98 (m, 1H), 1.30 (d, J = 6.8 Hz, 6H); ¹³C NMR (101 MHz, CDCl₃) δ 195.1, 141.9, 140.2, 135.9, 132.1, 129.2, 129.2, 129.1, 128.0, 127.7, 127.6, 127.3, 36.5, 35.7, 23.1; IR (liquid film, cm⁻¹) 3061, 3032, 2961, 2925, 2866, 1675, 1598, 1584, 1478, 1453, 1421, 1383, 1307, 1245, 1179, 1156, 1053, 1033, 1015, 905, 813, 752, 698, 614; HRMS (EI): m/z calcd. For $C_{17}H_{18}OS$ [M]⁺: 270.1073, found: 270.1073.

1-(4-(dimethylamino)phenyl)-2-(isopropylthio)ethan-1-one (5ha). Colorless oil, 105.1 mg, 89% yield; ¹H NMR (400 MHz, CDCl₃) δ 7.89 (d, *J* = 8.8 Hz, 2H), 6.63 (d, *J* = 7.2 Hz, 2H), 3.74 (s, 3H), 3.08–2.95 (m, 7H), 1.26 (d, *J* = 6.8 Hz, 6H); ¹³C NMR (101 MHz, CDCl₃) δ 193.5, 153.5, 131.1, 123.2, 110.7, 40.1, 36.0, 35.5, 23.1; IR (liquid film, cm⁻¹) 3061, 3032, 2961, 2925, 2866, 1675, 1598, 1584, 1478, 1453, 1421, 1366, 1307, 1245, 1179, 1156, 1053, 1033, 1015, 905, 813, 752, 698, 614; HRMS (EI): *m/z* calcd. For C₁₃H₁₉NOS [M]⁺: 237.1182, found: 237.1184.

2-(isopropylthio)-1-(3-phenoxyphenyl)ethan-1-one (5ia). Pale pink oil, 97.8 mg, 68% yield; ¹H NMR (400 MHz, CDCl₃) δ 7.69 (d, *J* = 7.6 Hz, 1H), 7.61 (s, 1H), 7.44–7.40 (m, 1H), 7.38–7.34 (m, 2H), 7.22–7.19 (m, 1H), 7.16–7.13 (m, 1H), 7.04–7.02 (m, 2H), 3.78 (s, 2H), 3.02–2.92 (m, 1H), 1.27 (d, *J* = 6.8 Hz, 6H); ¹³C NMR (101 MHz, CDCl₃) δ 194.5, 157.9, 156.6, 137.2, 130.11, 130.08, 124.0, 123.6, 123.5, 119.3, 118.6, 36.5, 35.6, 23.0; IR (liquid film, cm⁻¹) 3061, 3032, 2961, 2925, 2866, 1675, 1598, 1584, 1478, 1453, 1421, 1366, 1307, 1245, 1179, 1156, 1053, 1033, 1015, 905, 813, 752, 698, 614; HRMS (EI): *m/z* calcd. For C₁₇H₁₈O₂S [M]⁺: 286.1022, found: 286.1027.

2-(isopropylthio)-1-(4-methoxyphenyl)ethan-1-one (5ja). Colorless oil, 93.2 mg, 82% yield; ¹H NMR (400 MHz, CDCl₃) δ 7.95 (d, *J* = 8.8 Hz, 2H), 6.93 (d, *J* = 8.8 Hz, 2H), 3.86 (s, 3H), 3.77 (s, 2H), 3.03–2.93 (m, 1H), 1.27 (d, *J* = 6.8 Hz, 6H); ¹³C NMR (101 MHz, CDCl₃) δ 193.9, 163.7, 131.2, 128.4, 113.9, 55.6, 36.2, 35.6, 23.1; IR (liquid film, cm⁻¹) 3061, 3032, 2961, 2925, 2866, 1675, 1598, 1584, 1478, 1453, 1421, 1383, 1366,

1307, 1245, 1179, 1156, 1137, 1053, 1015, 752, 698, 614; HRMS (EI): *m/z* calcd. For C₁₂H₁₆O₂S [M]⁺: 224.0866, found: 224.0865.

2-(isopropylthio)-1-(3-methoxyphenyl)ethan-1-one (5ka). Colorless oil, 89.5 mg, 79% yield; ¹H NMR (400 MHz, CDCl₃) δ 7.54–7.52 (m, 1H), 7.50 (s, 1H), 7.38–7.34 (m, 1H), 7.12–7.09 (m, 1H), 3.84 (s, 3H), 3.81 (s, 2H), 3.03–2.93 (m, 1H), 1.27 (d, *J* = 6.8 Hz, 6H); ¹³C NMR (101 MHz, CDCl₃) δ 194.9, 159.9, 136.8, 129.7, 121.5, 119.9, 113.1, 55.5, 36.5, 35.6, 23.0; IR (liquid film, cm⁻¹) 3061, 3032, 2961, 2925, 2866, 1675, 1598, 1584, 1478, 1453, 1421, 1383, 1366, 1307, 1245, 1179, 1156, 1137, 1053, 1033, 1015, 999.9, 905, 752, 698, 614; HRMS (EI): *m/z* calcd. For C₁₂H₁₆O₂S [M]⁺: 224.0866, found: 224.0865.

1-(3,4-dimethoxyphenyl)-2-(isopropylthio)ethan-1-one (5la). Colorless oil, 97.7 mg, 78% yield; ¹H NMR (400 MHz, CDCl₃) δ 7.58 (dd, *J* = 8.4, 2.0 Hz, 1H), 7.53 (d, *J* = 2.0 Hz, 1H), 6.87 (d, *J* = 8.4 Hz, 1H), 3.93 (s, 3H), 3.92 (s, 3H), 3.77 (s, 2H), 3.04–2.94 (m, 1H), 1.27 (d, *J* = 6.8 Hz, 6H); ¹³C NMR (101 MHz, CDCl₃) δ 194.0, 153.3, 149.2, 128.5, 123.7, 110.8, 110.1, 56.2, 56.1, 36.1, 35.7, 23.1; IR (liquid film, cm⁻¹) 3061, 3032, 2961, 2925, 2866, 1675, 1598, 1584, 1478, 1453, 1421, 1383, 1366, 1307, 1245, 1179, 1156, 1137, 1033, 1015, 905, 813, 752, 698, 614; HRMS (EI): *m/z* calcd. For C₁₃H₁₈O₃S [M]⁺: 254.0971, found: 254.0976.

1-(3,4-dimethoxyphenyl)-2-(isopropylthio)ethan-1-one (5ma). Pale pink oil, 84.8 mg, 70% yield; $^1\text{H NMR}$ (400 MHz, CDCl_3) δ 8.51 (s, 1H), 8.03 (d, $J = 8.4$ Hz, 1H), 7.96 (d, $J = 8.0$ Hz, 1H), 7.91-7.86 (m, 2H), 7.62-7.53 (m, 2H), 3.95 (s, 2H), 3.08-2.99 (m, 1H), 1.30 (d, $J = 6.8$ Hz, 6H); $^{13}\text{C NMR}$ (101 MHz, CDCl_3) δ 195.1, 135.7, 132.7, 132.6, 130.7, 129.8, 128.7, 128.6, 127.9, 126.9, 124.5, 36.5, 35.7, 23.1; IR (liquid film, cm^{-1}) 3061, 3032, 2961, 2925, 2866, 1675, 1598, 1584, 1478, 1453, 1421, 1383, 1307, 1245, 1179, 1156, 1053, 1033, 1015, 999.9, 752, 698, 614; HRMS (EI): m/z calcd. For $\text{C}_{15}\text{H}_{16}\text{OS}$ $[M]^+$: 244.0916, found: 244.0915.

mutual δ identical

4. Control Experiments

4.1 Capture of enolate intermediate

To a 25 mL Schlenk tube equipped with a Teflon septum and magnetic stir bar were added benzamide **1a** (102.7 mg, 0.50 mmol), methyl phenyl sulfide **2a** (82 μL , 0.70 mmol, 1.4 equiv), THF (1.0 mL) and LDA (58.9 mg, 0.55 mmol, 1.1 equiv). The tube was sealed and stirred at 40 °C for 24 hours. The reaction mixture was then cooled to room temperature. BnBr (148 μL , 1.25 mmol, 2.5 equiv) was added to the reaction mixture under nitrogen atmosphere. After stirring at 25 °C for another 24 hours, the reaction mixture was quenched by adding 10 mL of H_2O and then diluted by ethyl acetate (EtOAc, 30 mL). The aqueous solution was extracted thrice with EtOAc (30 mL x 3) and the combined organic layer was washed with brine (10 mL), dried over anhydrous Na_2SO_4 . After filtration and concentration by rotary evaporation, the residue was purified by silica gel column chromatography (PE/EtOAc = 50/1) to afford the desired product **6a** (119.7 mg, 0.38 mmol, 79% yield for three steps) as a white solid.

colorless

1,3-diphenyl-2-(phenylthio)propan-1-one (6a).¹² White solid, 125.4 mg, 79% yield, m.p. 78-81 °C; $^1\text{H NMR}$ (400 MHz, CDCl_3): δ 7.82 (d, $J = 7.6$ Hz, 2H), 7.51-7.47 (m, 1H), 7.38-7.34 (m, 2H), 7.31-7.16 (m, 10H), 4.69 (dd, $J = 8.4, 6.0$ Hz, 1H), 3.40 (dd, $J = 14.0, 8.4$ Hz, 1H), 3.13 (dd, $J = 14.0, 6.0$ Hz, 1H); $^{13}\text{C NMR}$ (101 MHz, CDCl_3) δ 195.3, 138.8, 136.3, 134.7, 133.1, 132.1, 129.4, 129.1, 128.9, 128.63, 128.60, 126.7, 53.0, 37.4; IR (KBr, cm^{-1}) 3071, 3059, 3022, 2964, 2919, 1668, 1593, 1579, 1493, 1447, 1438, 1432, 1362, 1246, 1180, 1171, 947, 747, 701, 691, 683, 659, 628, 515.

IC missing?

S26

4.2 Parallel Intermolecular Competitive Reaction

$$k_H/k_D = 0.4467/0.1419 = 3.2$$

To a 25 mL Schlenk tube equipped with a Teflon septum and magnetic stir bar were added benzamide **1a** (41.0 mg, 0.20 mmol), methyl sulfide **1a** (34.8 mg, 0.28 mmol), 1.4 equiv., *n*-dodecane (internal standard, 15.5 mg) and THF (4 mL). The reaction tube was sealed and placed in ice-water bath. Then a mixture of LDA (23.6 mg, 0.22 mmol, 1.1 equiv.) and THF (1 mL) was added to the reaction mixture under nitrogen atmosphere. The tube was then sealed and continuously stirred in ice-water bath. Data were sampled and analyzed by GC for every 5 minutes. The reaction of deuterated methyl sulfide **2a-d** was carried out and monitored similarly.

Figure S1. Plot of Yield (%) versus Time (s) for Kinetic Isotope Effect Experiment with Methyl Sulfide **2a** and Deuterated Methyl Sulfide **2a-d**.

S27

4.3 Intramolecular Competitive Reaction

To a 25 mL Schlenk tube equipped with a Teflon septum and magnetic stir bar were added benzamide **1a** (61.6 mg, 0.30 mmol), methyl sulfide **2n-d** (72.9 mg, 0.42 mmol, 1.4 equiv.), THF (5.0 mL) and LDA (35.3 mg, 0.33 mmol, 1.1 equiv.). The tube was sealed and stirred at 25 °C for 5 min. The mixture was then quenched by adding five drops of H₂O and then diluted by ethyl acetate (EtOAc, 30 mL). The aqueous solution was dried over anhydrous Na₂SO₄. After filtration and concentration by rotary evaporation, the residue was purified by silica gel column chromatography (PE/EtOAc = 50/1) to afford the desired product **3an-d** (36.0 mg, 0.13 mmol, 43% yield) as a colorless oil.

S28

4.4 Deuterium Scrambling Reaction 1

To a 25 mL Schlenk tube equipped with a Teflon septum and magnetic stir bar were added benzamide **1a** (102.7 mg, 0.50 mmol), deuterated methyl sulfide **2a-d** (89.1 mg, 0.70 mmol, 1.4 equiv.), THF (1.0 mL) and LDA (58.9 mg, 0.55 mmol, 1.1 equiv.). The tube was sealed and stirred at 40 °C for 24 hours. The mixture was then cooled to room temperature, quenched by adding five drops of H₂O and then diluted by ethyl acetate (EtOAc, 30 mL). The aqueous solution was dried over anhydrous Na₂SO₄. After filtration and concentration by rotary evaporation, the residue was purified by silica gel column chromatography (PE/EtOAc = 50/1) to afford the desired product **3aa-d** (86.7 mg, 0.38 mmol, 76% yield) as a colorless oil.

S29

4.5 Deuterium Scrambling Reaction 2 formation tertiary alcohol?

To a 25 mL Schlenk tube equipped with a Teflon septum and magnetic stir bar were added non-deuterated ketone **3aa** (112.8 mg, 0.49 mmol), deuterated methyl phenyl sulfide **2a-d** (89.1 mg, 0.70 mmol, 1.4 equiv.), THF (1.0 mL) and LDA (58.9 mg, 0.55 mmol, 1.1 equiv.). The tube was sealed and stirred at 40 °C for 24 hours. The mixture was then cooled to room temperature, quenched by adding five drops of H₂O and then diluted by ethyl acetate (EtOAc, 30 mL). The aqueous solution was dried over anhydrous Na₂SO₄. After filtration and concentration by rotary evaporation, the residue was purified by silica gel column chromatography (PE/EtOAc = 50/1) to isolated the compound of ketone (38.8 mg, 0.21 mmol, 42% recovered) as a colorless oil. ¹H NMR analysis of the isolated ketone revealed that no deuterium atoms were incorporated in the isolated ketone. The above result indicated the C-H/C-D exchange in our reaction occurs before the C-C bond formation process.

S30

Dear reviewers,

Thank you very much for your comments and suggestions about the manuscript entitled “Direct Alkylation of Unactivated Benzamides with Methyl Sulfides by Breaking Both C-N and C-H Bonds under Transition-Metal-Free Condition (COMMSCHEM-21-0086-T)”. Your valuable comments that have undoubtedly improved the quality of the manuscript. In the light of the comments, we revised the manuscript and provide responses to every single questions one by one. I hope the responses are in line with your comments, and would clarify the concerns.

Best regards,
Sincerely yours,

Bing-Tao Guan

Comments and responses:

Reviewer #1 (Remarks to the Author):

Comment 1.1:

The scope to of the aryl groups in benzamide derivatives and of methyl sulfides were examined. However, that of N-substituents was not mentioned.

Response 1.1:

We carried out the reactions of several tertiary benzamides and the results were added in the SI (Table S3). N,N-Dimethyl, N,N-Diethyl, piperidinyl and piperidinyl benzemides could undergo the benzylation reaction with thioanisole but gave the product in low yields of 26-45%. N-(tert-butyl)-N-ethylbenzamide gave a moderate yield of 69%. The steric bulky N,N-diisopropyl and N,N-dicyclohexyl benzamides afforded the product in high yields of 93-95%. A comment about these results was added in the text.

Comment 1.2: *Thus the method is narrow in scope, restricting to N,N-diisopropylaroylamides and methyl sulfides as the starting materials and α -sulfonylated aromatic carbonyl compounds as the products. Even for the same class of products, Pace’s method (EJOC 2018, 2466, cited as ref. 77) is larger in scope.*

Response 1.2:

For the scope of methyl sulfides, we actually don’t regard it as a limitation. On the contrary, we think the reaction of methyl sulfides displays excellent chemoselectivity of this reaction. For the acidity of α -C-H bond of alkyl sulfides and the coordinating ability of the sulfide group, there are several C-H bond including α -C-H bonds of alkyl sulfides and *ortho*-C-H bond of thioanisoles that could possibly be involved in the reaction. However, the reaction selectively took place on the methyl group of the methyl sulfides that revealing the excellent chemoselectivity.

For the scope of the α -sulfenylated ketone products, we agree that Pace did a nice work for the synthesis of α -sulfenylated ketones from Weinreb amides and α -sulfur methyllithium reagents. Even discounting the use of Weinreb amides, 2 equiv. of BuLi, 2 equiv. of DABCO or 2 equiv. of ClCH₂SMe and 7 equiv. of Li metal, Pace showed us only 5 sulfide nucleophiles and total 18 examples of α -sulfenylated ketone products. I really could not agree that “Pace’s method is larger in scope”. Maybe the referee only means the scope of the amides. But we think it is unfair to us to only compare the scope of Weinreb amides with our work. Besides, throughout the manuscript, we never claimed any advantage over the scope of this reaction.

Comment 1.3:

In the title and conclusion section, by what means “Unactivated Benzamides”?

Response 1.3:

The stability of the amide group mostly comes from the resonance between nitrogen atom and the carbonyl group. Many approaches including EWG groups on N atom, distorted structures and addition of a strong electrophilic agent such as triflic anhydride were considered as activation effects, which is of help to lower the resonance structure. The amides without those factors were thus called “unactivated amides”. Morimoto and Takashi Ohshima used “unactivated amides” and “unactivated amide bond” from 2012 (Angew. Chem. Int. Ed. 2012, 51, 8564–8567; Chem. Commun., 2014, 50, 12623–12625; Org. Process Res. Dev. 2019, 23, 4, 588–594). Sozastak and Procter used “unactivated amides” in 2013 (Angew. Chem. Int. Ed. 2013, 52, 7237–7241; J. Am. Chem. Soc. 2014, 136, 2268–2271; J. Am. Chem. Soc. 2019, 141, 28, 11161–11172). In the review “Transition-Metal-Catalyzed Cleavage of C–N Single Bonds” (Chem. Rev. 2015, 115, 12045–12090), Prof. Xi used “unactivated C–N bond” of amides. Kim and Lee recently used “Unactivated tertiary benzamides” in 2020 (Org. Biomol. Chem., 2020, 18, 6053–6057). To describe the substrates more accurately, we revised the “Unactivated Benzamides” in the title and abstract as “unactivated tertiary benzamides”.

Comment 1.4:

The authors seem to lack a comprehensive understanding of the current status of the amide chemistry. Consequently, the first and second sentences and other sentences in the abstract are inappropriate. A fast and general route to ketones from amides and organolithium compounds under aerobic conditions appeared recently: Chem.–Eur. J. 2021, 27, 2868–2874. “Without using transition-metal catalysts, organometallic reagents or strong bases”, the coupling of amides with alkenes to give ketones or enones are known (Chin. J. Chem. 2019, 37, 887; Chin. J. Chem. 2019, 37, 811; Chin. J. Chem. 2019, 37, 315 and cited references). On the other hand, the preparations of ketones from N,N-dimethylamides and N,N-diethylamides are known: Synthesis 1984, 228 (cited as ref. 6) and J. Org. Chem. 1986, 51, 3566.

Response 1.4:

Thanks for your kind reminding. We feel embarrassed that we missed some of the most interesting and important developments. The work (Chem.–Eur. J. 2021, 27, 2868–2874.) was added as ref. 27 and a comment was added in the text. The work (J. Org. Chem. 1986, 51, 3566.) was cited as ref.

7. A comment about the couplings of amides with alkenes was added in the text, and the papers mentioned by the referee were cited as ref. 58-65.

The referee stated that “Without using transition-metal catalysts, organometallic reagents or strong bases, the coupling of amides with alkenes to give ketones or enones are known”, On the other hand, the preparations of ketones from N,N-dimethylamides and N,N-diethylamides are known”. We commented about the reactivity of amide in the abstract as that “Amide, a fundamental and widespread functional group, is usually considered as a poor electrophile for the resonance stability of the amide bond. Various approaches have been developed to address the challenges in the amide transformations. Nonetheless, most had to use activated amides, organometallic reagents or transition metal catalysts.” We just commented about our work “This approach successfully achieved an efficient and selective synthesis of α -sulfenylated ketones without using transition-metal catalysts, organometallic reagents or strong bases.” We neither claimed any excessive statement nor denied the known methods. We don't think there were any expression inappropriate in the abstract. So, we tend to stick to most of the statements.

Comment 1.5:

Another key issue is about the definition of amides. Fig. 1, a represents the widely accepted definition of amides (herein, it is better changing Ph to R). However, in the text including Fig. 1, b (middle), and many references, different kind of diacyl, and triacylamines were also called amides. This is problematic, because diacyl, and triacylamines have not resonance structures shown in Fig. 1, a, namely, the (first) carbonyl group in such so called “amides” is as reactive as a ketone carbonyl. As such the cited references 27-52 and related text should be deleted.

Response 1.5:

Thanks for the comments and suggestions. In Fig. 1, we have changed the Ph to R. For the definition of amides, we strongly agree with the referee that some of the activated amides don't have any characters of general amides. However, most of them were called amides in many papers. It is hard for us to argue that with the published papers. But considering this suggestion, we selectively delete the references 32-52.

Comment 1.6:

A reference on the ortho-metalation with LDA should be cited: Angew. Chem. Int. Ed. 2019, 58, 7313.

Response 1.6: Thanks for the comments. This reference (Angew. Chem. Int. Ed. 2019, 58, 7313) is cited in the revised manuscript as reference 76.

Comment 1.7:

In the SI general protocol, please indicate the LDA used, solid or a solution?

Response 1.7:

We used LDA in solid in a glove box for convenience. The commercial LDA solutions (2.0 M or 1.0 M in THF/n-heptane/ethylbenzene) purchased from several companies display comparable reactivity with the solid LDA. A new table (table S4) with behaviors of different LDA was added in the SI.

Comment 1.8:

In the SI, for the ¹³C NMR data, the peak numbers didn't consist with the structure.

Response 1.8:

We have carefully rechecked all of the ¹³C NMR data of the products. We have compared the spectra acquired with the literature spectra of the reported compounds. There are no mistakes about the ¹³C NMR data. The inconsistencies between the peak number of ¹³C NMR and the structure could occur when some of the aromatic carbons have very close chemical shifts, which could overlap sometimes.

Reviewer #2 (Remarks to the Author):

Comment 2.1:

The authors are supposed to remove any ambiguous description, such as “relative weak base” for highlighting the LDA, which is commonly regarded as strong base with bulky structure.

Response 2.1:

Thanks for the comments. We believe that LDA is commonly regarded as strong base. Thus, we avoid general comments “weak base” about LDA and use “relatively weak base” when taking about the deprotonation of thioanisole in the revised manuscript.

Comment 2.2:

In terms of benzamides, dimethyl groups on N are more generally utilized to represent the unactuated benzamide. However, the authors only exemplified the i-Pr. Accordingly, i-Pr is more likely to facilitate the reaction. The authors are recommended to give both experiments and discussions regarding the i-Pr.

Response 2.2: Thanks for the comments. Please refer to response 1.1.

Comment 2.3:

In the mechanistic discussion, the authors concluded that from A to B is the rate-determining step without related evidence. The authors are recommended to provide the DFT calculation on the proposed transition state. Or at least, the provable literatures should be referred.

Response 2.3: Thanks for your suggestion. Most deprotonation of thioanisole reaction used BuLi in the presence of DABCO or TMEDA. The deprotonation of thioanisole with PhLi could be found twice (Chemische Berichte (1985), 118, (6), 2330-42; Bailey, S. (2001). Thioanisole. In Encyclopedia of Reagents for Organic Synthesis, (Ed.)). There is no similar process with our deprotonation of thioanisole with ortho-lithium phenyl compound reported. The ortho-lithium benzamide complex was prepared according reported procedure. The reaction between ortho-lithium benzamide complex and thioanisole smoothly afforded the ketone product in 98% yield, suggesting that the ortho-lithium benzamide complex could be the true intermediate in the reaction (SI, 3.7).

In the early version, we provide primary kinetic isotope effects (KIE) with 3.2 and 3.5 for intermolecular and intramolecular competitive reaction of thioanisole and deuterated thioanisole. The primary kinetic isotope effects are the evidence for our proposal that “the cleavage of the methyl C–H bond of thioanisole could be the rate determining step”. In addition, we did not

“concluded” that from A to B is the rate-determining step. We just provided a possibility. A DFT calculation could provide some energy data for the intermediates and transition states, so as to provide information for the mechanism. However, it is not easy for our group, a group focuses on experimental chemistry. We prefer to figure out the problems with experiments. We measured the initial rate and found it to be first order in the amide, first order in LDA, first order in thioanisole, and minus one order in HDA, suggesting an equilibrium and a following rate-determining process. These kinetic data were added into to SI (3.8) and a comment was added in the text.

Reviewer #3 (Remarks to the Author):

Comment 3.1:

p1/3/7: metal amides should be considered as strong Brønsted bases; if the base strength is mentioned it should be clearly put in relation to the pK_a value of the corresponding conjugate acid. Also, when pK_a values of the reagents are compared it is not useful at all to consider two different solvents (DMSO/THF); as such data are not comparable. Also, when using Lewis acidic entities (here Li^+) complexation to basic sites in the substrates can significantly increase the relevant C–H bond acidity, which should be taken into consideration.

Response 3.1:

We agree that LDA is commonly regarded as strong base. In the submitted version, we just meant to emphasize that LDA is a relatively weak base compared with the base such as alkyl lithium that needed in the deprotonation of sulfides. In the revised manuscript, we avoided general comments “weak base” about LDA and used “relatively weak base” only when taking about the deprotonation of thioanisole.

Thanks for your kind reminding about the pK_a values in two different solvents (DMSO/THF). We just listed the pK_a values of HAD, amide and thioanisole in THF (from J. Chem. Soc., Chem. Commun. 1983, 620-621; and J. Am. Chem. Soc. 1983, 105, 7790-7791.). We didn't find the pK_a value of methyl alkyl sulfides. But theoretically they are less acidic than thioanisoles. These data were measured the deprotonation equilibrium between arenes or thioanisole and the lithium amide using ^{13}C NMR. So, the Lewis acidity of the Li^+ has already been involved in the data. And we do agree with the referee that the Li^+ cation could play an important role in the deprotonation process.

Comment 3.2:

p2: there is a confusing statement of “decreased nucleophilicity” regarding amides; explain properly or change to “electrophilicity” (which would make sense).

Response 3.2:

Sorry for this mistake. We have corrected it to “decreased electrophilicity”.

Comment 3.3:

p2: better to add a suitable ref when mentioning the potential side-reactions.

Response 3.3:

We do have several reference (ref 5-7) near the next sentence. We moved these references to the

sentence mentioning the potential side-reactions.

Comment 3.4:

p2: "ketamine" is the wrong term; it must be "ketiminium".

Response 3.4:

Thanks for your reminding. We removed the comment about the ketiminium intermediate and added a new one about the nitrilium ion intermediates.

Comment 3.5:

p2 (F1): R' must be defined for the three distinct reaction pathways (b1, b2, c).

Response 3.5:

In all of the three reaction pathways, R' could be alkyl, benzyl or aryl nucleophiles. R' means carbon nucleophiles. So, we think it is better to leave it undefined.

Comment 3.6:

p3: add the term "Lewis", otherwise misleading.

Response 3.6:

Thanks for your kind reminding. We corrected it to "by using 1,1-diborylalkanes as pro-nucleophiles and alkyl lithium reagents as Lewis bases".

Comment 3.7:

p3: the relevant data from Liu's paper (ref 56) should be displayed in F1.

Response 3.7:

It is a nice suggestion to add Liu's work in the background of amides transformations (F1). We did carefully think of doing so in the preparation of the first manuscript. Liu's work is very interesting and displayed several reaction pathways. It is a little bit complex than what we can show in a limited space in figure 1. So, we preferred to give a comment in the text without a visual picture in figures.

Comment 3.8:

p3: the stabilization of the α -anion of S-containing compounds (hyperconjugation) must be added (plus a suitable ref).

Response 3.8:

Thanks for your kind reminding. A comment about the polarizability effect and hyperconjugation effect of sulfur atom was added in the text when talking about the acidity of the sulfides. Two papers about the increased acidity of sulfides were cited as ref. 58 and 59.

Comment 3.9:

p3: self-citation (refs 81–83) is OK but then relevant other studies (use of alkali metal amides in

C–H bond activation; all equilibria) must be included as well ... e.g. Kobayashi, Schneider;

Walsh, ...

Response 3.9:

We didn't cite our own papers only for self-citation. The paper cited is close about our idea that a relatively weak base could work even better than a strong one. These works are the logic basis of the present work. Prof. Kobayashi, Schneider, Walsh and others have done very excellent works in the field of alkali metal amides-catalyzed C-H bond activation. However, seldom of them pay attention to the "weak base" and the "deprotonation equilibrium". Prof. Walsh mentioned "equilibrium" twice. But those works are not so close to this work. So, we didn't cite them in this manuscript.

Comment 3.10:

p4/6: the deprotonation/quenching of thioanisole with LDA and LTMP –in the absence of the benzamide– must be carried under the reaction conditions of T1 (e.g. THF, 60 °C; then TMSCl) before stating it is unsuccessful (due to pK_a values that are not comparable anyway; as reported in different solvents). Also, if the bis-ortho-methylated benzamide (not a Brønsted acid) is used as a Lewis basic additive (to enhance the Brønsted basicity of LDA), what would be the outcome when mixing thioanisole and LDA?

Response 3.10:

The pK_a values of HAD, HTMP and thioanisole in THF are 35.7, 37.3 and 38.3, respectively. We also carried out the reaction of 4-phenylthioanisole with LDA and LTMP in THF at 60 °C for 24 h and then quenched by TMSCl. No TMS substituted 4-phenylthioanisole was detected, suggesting the deprotonation reaction of 4-phenylthioanisole by LDA and LTMP failed.

Also, treating 4-phenylthioanisole with LDA in the presence of bis-ortho-methylated benzamide at 60 °C for 24 h followed being quenched by TMSCl gave no TMS substituted 4-phenylthioanisole. More than 95% of the 4-phenylthioanisole was recovered.

Comment 3.11:

p4: there should be also "steric hindrance" included in the base/nucleophile discussion (i.e., bulky amides vs. slim alkyl reagents); it is completely missing.

Response 3.11:

Thanks for your kind suggestion. The steric hindrance, the basicity and the nucleophilicity and their relationship is interesting. We added a comment in the discussion about table 1 entries 3,4 VS 5,6.

Comment 3.12:

p4/6: are tertiary alcohols obtained at any stage in any of the conducted experiments (particularly for the alkyl lithium reagents and/or in the final control experiment)?

Response 3.12:

The reaction between benzamide and methyl sulfides using n-BuLi as base give tertiary alcohols as by-products. Notes about the tertiary alcohols were added in Table 1, Table S2. A detail explanation about the by-products was added in Figure S1 in SI.

Comment 3.13:

p4: regarding the amount of LDA used, refer to the SI (TS-1).

Response 3.13:

The amount of LDA was demonstrated clearly in Table S1 in the revised SI. The comment was referred to Table S1.

Comment 3.14:

p4: in a similar context, clarify the following points:

What is the outcome if Me-SPh (→ stoichiometric LDA; secondary center created) is replaced:

(a) by Et-SPh or TMSCH₂-SPh (→ stoichiometric LDA; tertiary center created) ?

(b) by iPr-SPh (→ potentially catalytic LDA; quaternary center created) ?

[may require more harsh conditions; dioxane, 80+ °C]

Response 3.14:

We tried the reaction between Et-SPh or *iPr*-SPh and tertiary benzamide in 1,4-dioxane under 80 °C and no desired benzoylated product was detected. In the both cases, the tertiary benzamide was recovered with more than 90% recovery. The reaction displays excellent selectivity to methyl C-H bond possibly because of the four-membered ring transition state structure. The steric hindrance prevents the formation of the highly charged transition state.

Comment 3.15:

p4/5 (T2/3): regarding the scope (which is in principle very good), what is the outcome in case of methyl sulfides or tertiary amides bearing an Ar group with an EWG (there is not any example in both Tables)?

Response 3.15:

We tried the reactions between tertiary amide with methyl sulfides bearing an Ar group with 4-CN, 4-NO₂ and 4-CO₂Me, respectively. However, none benzoylated products were detected and tertiary amide were recovered with more than 90% recovery in the three cases. We believe that the above failed examples were due to the incompatibility between the EWGs and LDA in our reaction system. For tertiary amide, those bearing 4-CN, 4-Bz, 4-F and 3-F (not bearing an Ar with 4-CN, 4-Bz, 4-F and 3-F) on the phenyl ring also failed to deliver the corresponding products. Besides, we found that 1.1 equiv. of tert-butyl benzoate used as additive could inhibit the reaction between tertiary amide and thioanisole promoted by LDA in THF under 40 °C dramatically. We guessed that additional coordination group would possibly catch the LDA and prevent the following process.

Comment 3.16:

p4/5 (T2/3): adjust structural display of “Bu” and “iPr” in the Tables.

Response 3.16:

The structural display of “Bu” and “iPr” in the Table 2 and Table 3 have been adjusted.

Comment 3.17:

Amide scope: exclusive use of aromatic tertiary N-isopropyl benzamides (= drawback of this methodology): what is the scope of the NR₂ portion? For instance, how about NEt₂, N(TMS)₂,

N(pyrrolidinyl)?

Response 3.17: Please refer to response 1.1.

Comment 3.18:

*p6/7 (F2/3): the control experiments conducted are reasonable (particularly the deuterium labelling ones) and support the proposed mechanistic pathway. It would be good though to try detecting postulated reaction intermediates (e.g. **A** and **D**) by mild HRMS methods (obviously in the absence of the sulfide and water, respectively).*

Response 3.18:

For the pK_a values of HAD and thioanisole in THF are 35.7 and 38.3, it is not easy to find **A** in the deprotonation equilibrium. We synthesized the ortho-lithiation intermediate according to the reported method (Angew. Chem. Int. Ed. 2001, 40, 1238). The complex exists in its dimeric structure $[A]_2$. Treating $[A]_2$ with 2.8 equiv. of thioanisole in THF at 25 °C delivered benzoylated product **3aa** with almost quantitative yield (98%) (SI, 3.7).

The quench of the reaction with BnBr gave further α -benzylated product **6a**. These control experiments could be regarded as the evidence of intermediate **A** and **D**.

Comment 3.19:

Sulfide deprotonation: include a suitable ref of a relevant Ar-Li-assisted “ σ -bond metathesis-type”

C-H bond metalation.

Response 3.19:

Please refer to response 2.3

Comment 3.20:

TOC: might be more reader-friendly to move “Control Experiments” forward (S-1).

Response 3.20:

“Control Experiments” was moved forward to the third part in SI.

Comment 3.21:

General Information: clarify the LDA confusion; add information on mp measurement (S-2).

Response 3.21:

For the LDA confusion, refer to response 1.7. An information about the mp measurement was added at the end of general information.

Comment 3.22:

General Procedures: better to add “m” [in mg] of methyl sulfide reagents, not just show “V” [in μ L] (S-4, S-26).

Response 3.22:

The mass information was added into the general procedures and Table S1.

Comment 3.23:

*Analytical Data: there seem to be accuracy issues with NMR data (almost on every single page ... S-6 ~ S-26); must be addressed; use the term “colorless” (not “white”) solid; some mp's look strange in terms of significant numbers (S-10, S-11, S-22). NMR Spectra: these display indeed sufficient purity of the isolated products; one exception though: based on the ¹H NMR chart, product **3ea** seems not pure (1.2~1.3 ppm); better to repurify and up-date yield.*

Response 3.23:

Many thanks for your kind careful review. We corrected the mistakes as pointed out. Compound **3ea** was repurified and the yield was up-dated. We have rechecked the ¹³C NMR data for all of the acquired compounds. For reported compounds, we have compared the spectra acquired with the literature spectra. For example, in the ¹³C NMR spectra of compounds **3aj** and **5ah** showed in the literatures, the peak numbers didn't consist with the structures either. This is mainly due to the overlap of peaks for the very closed chemical shifts.

Comment 3.24:

Reaction Optimization: clarify the LDA point; comment on the possibility of tertiary alcohol formation through nucleophilic addition in case of alkyl lithium species (S-3).

Response 3.24:

For the LDA point, see response 1.7; for the tertiary alcohol formation, see response 3.11.

Comment 3.25:

Control experiments: clarify two points (S-28, S-30).

Response 3.25:

In S-28, **3an-d** was obtained with 11% deuterium incorporation. No tertiary alcohol was detected in the reaction of ketone **3aa** with thioanisole **2a-d** in the presence of LDA.

REVIEWERS' COMMENTS:

Reviewer #1 (Remarks to the Author):

Although most (but not all) of the concerns of this reviewer have been addressed, the revisions and some responses are unsatisfied. The major problems being the definition of an amide and behave of an author of Nature Commun. should have on this issue. First, in Fig. 1, the authors have given a text book definition. As ref. 16 (JOC, 1989, 228) showed, the carbonyl groups in N-Ts, N-Cbz and N-Boc amides (are in fact some kind of imides) are as reactive as ketone carbonyls. In such situation, the addition of Grignard reagents is no longer challenging, but easy and high yielding (cf. ref. 15-18). The difference in reactivity between an amide and an imide is just as that between an ester and an anhydride. When citing a reference, the authors should first read the paper to get the scientific content instead of just based on the title of the paper, because many titles are misleading. Nature Commun., as a serious scientific journal with very high reputation within the scientific community, should try to tell readers the real science instead of misleading readers because of other journal To be concise, this reviewer insist that:

1. "Unactivated" should be removed from the title and from throughout the manuscript.
2. Ref. 33 should be deleted because it involve N,N-Di-Boc of primary amides. It is too much to be considered as amides.
3. The response for not alternate the abstract in not convincing. It should be revised to approach the reality and common sense.

Other required revisions:

1. Please re-cite the following two deleted references (ref. 36 and 41 in the previous version) because they involve amides:

36. Huang, P. & Chen, H. Ni-Catalyzed cross-coupling reactions of N-acylpyrrole-type amides with organoboron reagents. Chem. Commun. 53, 12584-12587 (2017).

41. Meng, G., Szostak, R. & Szostak, M. Suzuki–Miyaura Cross-Coupling of N-Acylpyrroles and Pyrazoles: Planar, Electronically Activated Amides in Catalytic N–C Cleavage. Org. Lett. 19, 3596-3599 (2017).

2. In the text, "Huang and Charette" should be revised as "Charette and Huang".
3. In the supporting information, please check the ¹³C NMR data to ensure that the carbon numbers are in consist with the molecular formula (e.g. 3a1).
3. In the supporting information, please check the ¹C NMR data to ensure the correctness (e.g. 3am: it lacks the data for two Me).
4. Please correct the style for all IUPAC names of compounds: The first latter in capital style, "tert" in italic style, etc.
5. In the section: "3.2 A general procedure for the alkylation of benzamide with *i*-PrSMe": how is possible: "The aqueous solution was dried over anhydrous Na₂SO₄."? Please check all.
6. In the section: "3.3 A gram-scale process for the alkylation of benzamide with thioanisole": "The resulting mixture was extracted thrice with"?

In summary, the revisions are far from satisfied, and major revisions are required.

Reviewer #2 (Remarks to the Author):

The authors have tackled all comments made by the referee. The manuscript is recommended to be accepted.

Reviewer #3 (Remarks to the Author):

I have carefully checked the revision responses as well as revised manuscript and SI. See below my final comments.

Comment 1.1:

The scope to of the aryl groups in benzamide derivatives and of methyl sulfides were examined. However, that of N-substituents was not mentioned.

Response 1.1:

We carried out the reactions of several tertiary benzamides and the results were added in the SI (Table S3). N,N-Dimethyl, N,N-Diethyl, piperidinyl and piperidinyl benzamides could undergo the benzylation reaction with thioanisole but gave the product in low yields of 26-45%. N-(tert-butyl)-N-ethylbenzamide gave a moderate yield of 69%. The steric bulky N,N-diisopropyl and N,N-dicyclohexyl benzamides afforded the product in high yields of 93-95%. A comment about these results was added in the text.

OK, addressed; refer also to 2.2. and 3.17.

However, please indicate in Table S3 –for the cases with 0–69% NMR yield– what the “other material” is: unreacted amide 1 and/or (un)defined side-products ?

Comment 1.2: Thus the method is narrow in scope, restricting to N,N-diisopropylaroylamides and methyl sulfides as the starting materials and α -sulfenylated aromatic carbonyl compounds as the products. Even for the same class of products, Pace’s method (EJOC 2018, 2466, cited as ref. 77) is larger in scope.

Response 1.2:

For the scope of methyl sulfides, we actually don’t regard it as a limitation. On the contrary, we think the reaction of methyl sulfides displays excellent chemoselectivity of this reaction. For the acidity of α -C-H bond of alkyl sulfides and the coordinating ability of the sulfide group, there are several C-H bond including α -C-H bonds of alkyl sulfides and ortho-C-H bond of thioanisoles that could possibly be involved in the reaction. However, the reaction selectively took place on the methyl group of the methyl sulfides that revealing the excellent chemoselectivity.

For the scope of the α -sulfenylated ketone products, we agree that Pace did a nice work for the synthesis of α -sulfenylated ketones from Weinreb amides and α -sulfur methyl lithium reagents. Even discounting the use of Weinreb amides, 2 equiv. of BuLi, 2 equiv. of DABCO or 2 equiv. of ClCH₂SMe and 7 equiv. of Li metal, Pace showed us only 5 sulfide nucleophiles and total 18 examples of α -sulfenylated ketone products. I really could not agree that “Pace’s method is larger in scope”. Maybe the referee only means the scope of the amides. But we think it is unfair to us to only compare the scope of Weinreb amides with our work. Besides, throughout the manuscript, we never claimed any advantage over the scope of this reaction.

Regarding the emphasized “chemoselectivity”: no surprise, it is expected. I would say that the scope reported by Pace is broader in terms of tertiary amides (aromatic, alkenyl, aliphatic), but the present submission offers a broader scope in terms of methyl sulfides (despite the lack of α -substituted substrates). The scope is less important anyway because the key points of the present submission are the more sustainable reaction conditions and the intriguing mechanistic picture. I think the scope has been sufficiently addressed with the additional examples presented in 1.1 / 3.17.

Comment 1.3:

In the title and conclusion section, by what means “Unactivated Benzamides”?

Response 1.3:

The stability of the amide group mostly comes from the resonance between nitrogen atom and the carbonyl group. Many approaches including EWG groups on N atom, distorted structures and addition of a strong electrophilic agent such as triflic anhydride were considered as activation effects, which is of help to lower the resonance structure. The amides without those factors were thus called “unactivated amides”. Morimoto and Takashi Ohshima used “unactivated amides” and “unactivated amide bond” from 2012 (*Angew. Chem. Int. Ed.* 2012, 51, 8564–8567; *Chem. Commun.*, 2014, 50, 12623–12625; *Org. Process Res. Dev.* 2019, 23, 4, 588–594). Sozastak and Procter used “unactivated amides” in 2013 (*Angew. Chem. Int. Ed.* 2013, 52, 7237–7241; *J. Am. Chem. Soc.* 2014, 136, 2268–2271; *J. Am. Chem. Soc.* 2019, 141, 28, 11161–11172). In the review “Transition-Metal-Catalyzed Cleavage of C–N Single Bonds” (*Chem. Rev.* 2015, 115, 12045–12090), Prof. Xi used “unactivated C–N bond” of amides. Kim and Lee recently used “Unactivated tertiary benzamides” in 2020 (*Org. Biomol. Chem.*, 2020, 18, 6053–6057). To describe the substrates more accurately, we revised the “Unactivated Benzamides” in the title and abstract as “unactivated tertiary benzamides”.

OK, the explanation and the term unactivated tertiary benzamides are fine – addressed.

Comment 1.4:

The authors seem to lack a comprehensive understanding of the current status of the amide chemistry. Consequently, the first and second sentences and other sentences in the abstract are inappropriate. A fast and general route to ketones from amides and organolithium compounds under aerobic conditions appeared recently: *Chem.–Eur. J.* 2021, 27, 2868–2874. “Without using transition-metal catalysts, organometallic reagents or strong bases”, the coupling of amides with alkenes to give ketones or enones are known (*Chin. J. Chem.* 2019, 37, 887; *Chin. J. Chem.* 2019, 37, 811; *Chin. J. Chem.* 2019, 37, 315 and cited references). On the other hand, the preparations of ketones from N,N-dimethylamides and N,N-diethylamides are known: *Synthesis* 1984, 228 (cited as ref. 6) and *J. Org. Chem.* 1986, 51, 3566.

Response 1.4:

Thanks for your kind reminding. We feel embarrassed that we missed some of the most interesting and important developments. The work (*Chem.–Eur. J.* 2021, 27, 2868–2874.) was added as ref. 27 and a comment was added in the text. The work (*J. Org. Chem.* 1986, 51, 3566.) was cited as ref. 7. A comment about the couplings of amides with alkenes was added in the text, and the papers mentioned by the referee were cited as ref. 58–65.

The referee stated that “Without using transition-metal catalysts, organometallic reagents or strong bases, the coupling of amides with alkenes to give ketones or enones are known”. On the other hand, the preparations of ketones from N,N-dimethylamides and N,N-diethylamides are known”. We commented about the reactivity of amide in the abstract as that “Amide, a fundamental and widespread functional group, is usually considered as a poor electrophile for the resonance stability of the amide bond. Various approaches have been developed to address the challenges in the amide transformations. Nonetheless, most had to use activated amides, organometallic reagents or transition metal catalysts.” We just commented about our work “This approach successfully achieved an efficient and selective synthesis of α -sulfenylated ketones without using transition-metal catalysts, organometallic reagents or strong bases.” We neither claimed any excessive statement nor denied the known methods. We don’t think there were any expression inappropriate

in the abstract. So, we tend to stick to most of the statements.

OK, addressed from my point of view. I assume the authors mean refs 37–44 (not refs 58–65), right?

Comment 1.5:

Another key issue is about the definition of amides. Fig. 1, a represents the widely accepted definition of amides (herein, it is better changing Ph to R). However, in the text including Fig. 1, b (middle), and many references, different kind of diacyl, and triacylamines were also called amides. This is problematic, because diacyl, and triacylamines have not resonance structures shown in Fig. 1, a, namely, the (first) carbonyl group in such so called “amides” is as reactive as a ketone carbonyl. As such the cited references 27-52 and related text should be deleted.

Response 1.5:

Thanks for the comments and suggestions. In Fig. 1, we have changed the Ph to R. For the definition of amides, we strongly agree with the referee that some of the activated amides don't have any characters of general amides. However, most of them were called amides in many papers. It is hard for us to argue that with the published papers. But considering this suggestion, we selectively delete the references 32-52.

OK, addressed from my point of view.

Comment 1.6:

A reference on the ortho-metalation with LDA should be cited: *Angew. Chem. Int. Ed.* 2019, 58, 7313.

Response 1.6: Thanks for the comments. This reference (*Angew. Chem. Int. Ed.* 2019, 58, 7313) is cited in the revised manuscript as reference 76.

OK, addressed. I assume that the authors mean ref 80 (not ref 76), right?

Comment 1.7:

In the SI general protocol, please indicate the LDA used, solid or a solution?

Response 1.7:

We used LDA in solid in a glove box for convenience. The commercial LDA solutions (2.0 M or 1.0 M in THF/n-heptane/ethylbenzene) purchased from several companies display comparable reactivity with the solid LDA. A new table (table S4) with behaviors of different LDA was added in the SI.

OK, addressed; refer also to 3.21 and 3.24.

Comment 1.8:

In the SI, for the ¹³C NMR data, the peak numbers didn't consist with the structure.

Response 1.8:

We have carefully rechecked all of the ¹³C NMR data of the products. We have compared the spectra acquired with the literature spectra of the reported compounds. There are no mistakes about the ¹³C NMR data. The inconsistencies between the peak number of ¹³C NMR and the structure could occur when some of the aromatic carbons have very close chemical shifts, which could overlap sometimes.

OK, addressed; refer also to 3.23.

Comment 2.1:

The authors are supposed to remove any ambiguous description, such as “relative weak base” for highlighting the LDA, which is commonly regarded as strong base with bulky structure.

Response 2.1:

Thanks for the comments. We believe that LDA is commonly regarded as strong base. Thus, we avoid general comments “weak base” about LDA and use “relatively weak base” when talking about the deprotonation of thioanisole in the revised manuscript.

The term “weak” should not appear at all in this context – it is misleading and inaccurate. For LDA/LTMP, please use the term “strong” base, or alternatively “moderately strong” base compared with alkyl lithium reagents (in the context of weakly acidic C–H bonds).

Comment 2.2:

In terms of benzamides, dimethyl groups on N are more generally utilized to represent the unactivated benzamide. However, the authors only exemplified the i-Pr. Accordingly, i-Pr is more likely to facilitate the reaction. The authors are recommended to give both experiments and discussions regarding the i-Pr.

Response 2.2: Thanks for the comments. Please refer to response 1.1.

OK, addressed; refer also to 1.1 and 3.17.

Comment 2.3:

In the mechanistic discussion, the authors concluded that from A to B is the rate-determining step without related evidence. The authors are recommended to provide the DFT calculation on the proposed transition state. Or at least, the provable literatures should be referred.

Response 2.3: Thanks for your suggestion. Most deprotonation of thioanisole reaction used BuLi in the presence of DABCO or TMEDA. The deprotonation of thioanisole with PhLi could be found twice (Chemische Berichte (1985), 118, (6), 2330-42; Bailey, S. (2001). Thioanisole. In Encyclopedia of Reagents for Organic Synthesis, (Ed.)). There is no similar process with our deprotonation of thioanisole with ortho-lithium phenyl compound reported. The ortho-lithium benzamide complex was prepared according reported procedure. The reaction between ortho-lithium benzamide complex and thioanisole smoothly afforded the ketone product in 98% yield, suggesting that the ortho-lithium benzamide complex could be the true intermediate in the reaction (SI, 3.7).

In the early version, we provide primary kinetic isotope effects (KIE) with 3.2 and 3.5 for intermolecular and intramolecular competitive reaction of thioanisole and deuterated thioanisole. The primary kinetic isotope effects are the evidence for our proposal that “the cleavage of the methyl C–H bond of thioanisole could be the rate determining step”. In addition, we did not “concluded” that from A to B is the rate-determining step. We just provided a possibility.

A DFT calculation could provide some energy data for the intermediates and transition states, so as to provide information for the mechanism. However, it is not easy for our group, a group focuses on experimental chemistry. We prefer to figure out the problems with experiments. We measured the initial rate and found it to be first order in the amide, first order in LDA, first order in thioanisole, and minus one order in HDA, suggesting an equilibrium and a following rate-determining process. These kinetic data were added into to SI (3.8) and a comment was added in the text.

OK, addressed; refer also to 3.19. Please add the refs mentioned above (thioanisole deprotonation by PhLi) in the mechanism discussion text portion.

Comment 3.1:

p1/3/7: metal amides should be considered as strong Brønsted bases; if the base strength is mentioned it should be clearly put in relation to the pKa value of the corresponding conjugate acid. Also, when pKa values of the reagents are compared it is not useful at all to consider two different solvents (DMSO/THF); as such data are not comparable. Also, when using Lewis acidic entities (here Li+) complexation to basic sites in the substrates can significantly increase the relevant C–H bond acidity, which should be taken into consideration.

Response 3.1:

We agree that LDA is commonly regarded as strong base. In the submitted version, we just meant to emphasize that LDA is a relatively weak base compared with the base such as alkyl lithium that needed in the deprotonation of sulfides. In the revised manuscript, we avoided general comments “weak base” about LDA and used “relatively weak base” only when talking about the deprotonation of thioanisole.

Thanks for your kind reminding about the pKa values in two different solvents (DMSO/THF). We just listed the pKa values of HAD, amide and thioanisole in THF (from J. Chem. Soc., Chem. Commun. 1983, 620-621; and J. Am. Chem. Soc. 1983, 105, 7790-7791.). We didn't find the pKa value of methyl alkyl sulfides. But theoretically they are less acidic than thioanisoles. These data were measured the deprotonation equilibrium between arenes or thioanisole and the lithium amide using ¹³C NMR. So, the Lewis acidity of the Li+ has already been involved in the data. And we do agree with the referee that the Li+ cation could play an important role in the deprotonation process.

The term “weak” should not appear at all in this context – it is misleading and inaccurate. For LDA/LTMP, please use the term “strong” base, or alternatively “moderately strong” base compared with alkyl lithium reagents (in the context of weakly acidic C–H bonds). It is appreciated that the pKa values are now “standardized” for THF as a common solvent. Fixed.

Comment 3.2:

p2: there is a confusing statement of “decreased nucleophilicity” regarding amides; explain properly or change to “electrophilicity” (which would make sense).

Response 3.2:

Sorry for this mistake. We have corrected it to “decreased electrophilicity”.

OK, addressed. However, please correct the term as well in Figure 1a (see annotated manuscript).

Comment 3.3:

p2: better to add a suitable ref when mentioning the potential side-reactions.

Response 3.3:

We do have several reference (ref 5-7) near the next sentence. We moved these references to the sentence mentioning the potential side-reactions.

OK, addressed.

Comment 3.4:

p2: “ketamine” is the wrong term; it must be “ketiminium”.

Response 3.4:

Thanks for your reminding. We removed the comment about the ketiminium intermediate and added a new one about the nitrilium ion intermediates.

OK, addressed.

Comment 3.5:

p2 (F1): R' must be defined for the three distinct reaction pathways (b1, b2, c).

Response 3.5:

In all of the three reaction pathways, R' could be alkyl, benzyl or aryl nucleophiles. R' means carbon nucleophiles. So, we think it is better to leave it undefined.

OK, addressed.

Comment 3.6:

p3: add the term "Lewis", otherwise misleading.

Response 3.6:

Thanks for your kind reminding. We corrected it to "by using 1,1-diborylalkanes as pro-nucleophiles and alkyl lithium reagents as Lewis bases".

OK, addressed.

Comment 3.7:

p3: the relevant data from Liu's paper (ref 56) should be displayed in F1.

Response 3.7:

It is a nice suggestion to add Liu's work in the background of amides transformations (F1). We did carefully think of doing so in the preparation of the first manuscript. Liu's work is very interesting and displayed several reaction pathways. It is a little bit complex than what we can show in a limited space in figure 1. So, we preferred to give a comment in the text without a visual picture in figures.

It would have been better to include (the part on tertiary benzamides that react with in situ-generated alkyl anions to form the corresponding ketones) for a more scholarly presentation (as it is very much related). If it is a matter of space, you could have removed Figure 1a (knowledge of undergrad level year 1). Anyway, I won't insist – the editor can decide.

Comment 3.8:

p3: the stabilization of the α -anion of S-containing compounds (hyperconjugation) must be added (plus a suitable ref).

Response 3.8:

Thanks for your kind reminding. A comment about the polarizability effect and hyperconjugation effect of sulfur atom was added in the text when talking about the acidity of the sulfides. Two papers about the increased acidity of sulfides were cited as ref. 58 and 59.

OK, addressed.

Comment 3.9:

p3: self-citation (refs 81–83) is OK but then relevant other studies (use of alkali metal amides in C–H bond activation; all equilibria) must be included as well ... e.g. Kobayashi, Schneider, Walsh, ...

Response 3.9:

We didn't cite our own papers only for self-citation. The paper cited is close about our idea that a relatively weak base could work even better than a strong one. These works are the logic basis of the

present work. Prof. Kobayashi, Schneider, Walsh and others have done very excellent works in the field of alkali metal amides-catalyzed C-H bond activation. However, seldom of them pay attention to the “weak base” and the “deprotonation equilibrium”. Prof. Walsh mentioned “equilibrium” twice. But those works are not so close to this work. So, we didn’t cite them in this manuscript.

I appreciate the authors’ standpoint, but I disagree. I had stated that self-citation (to a reasonable extent) is of course OK. However, relevant publications by other groups –particularly when these were published earlier and displaying alkali metal amide catalysis rather than the “stoichiometric” use of alkali metal amides, like in the present submission– must be respected and thus included. By the way, when the pKa difference (base vs. C–H acidic substrate) is e.g. +/- 3 units, a deprotonative equilibrium is expected – it does not have to be explicitly stated. Therefore, the following [adjusted] citations are expected (for accuracy/fairness – it does not take anything away from the present submission, but the reader must be more suitably informed):

(1) Initial (self)refs 81 and 83 correspond now to (self)refs 74 (ACIE 2018, 57, 1650) and 75 (OL 2019, 21, 5351); these are appropriate indeed (no issue at all). It is appreciated that the authors removed – on their own– initial self-ref 82 (ACIE 2018, 57, 8245): a beautiful study but not so relevant in the context of the present submission.

(2) The following (earlier) papers by other groups must be cited:

– JACS 2017, 139, 4362:

Published earlier than refs 74 and 75; catalytic use of Li/Na/K amide (Na = best) as a “moderate base” for the deprotonation of weakly acidic C–H bonds; give generally better results than MeLi (“strong base”; see Table 1); an equilibrium (deprotonation/reprotonation; due to the generation of a non-gaseous conjugate acid = secondary amine) has been clearly proposed (display of equilibrium arrow in Scheme 1b) and proved in both optimization experiments (Table 1: formation of isomerization side-product 1’a; plus text) and mechanism experiments (Scheme 4i: formation of isomerization side-product 1’a; plus text).

– ACIE 2018, 57, 6896:

Published earlier than ref 75; catalytic use of K amide (in the presence of LiOR) as a “moderate base” for the deprotonation of weakly acidic C–H bonds. Here, a deprotonative equilibrium is simply expected because of the small pKa difference – no need to mention.

– NatComm 2018, 9, 3365:

Published earlier than ref 75; stoichiometric use of Na/K amide (Na = best; in the presence of CsX) as a “moderate base” for the deprotonation of weakly acidic C–H bonds. Here, a deprotonative equilibrium has been clearly stated (display of equilibrium arrow between 2 and C in Figure 3; plus text).

In turn, may I ask to please add this refs ?

(3) The authors have included two new (self)refs 76 & 77 (and I am a little surprised that these have not been drawn clearly to the attention of the editors/reviewers):

– (Self)ref 76 = Sci. China. Chem. 51, 201 (2021):

This paper is a kind of account-style review by the authors. Of course, it is OK to be included as it refers to earlier contributions by the authors and other PIs (in this specific field of C–H bond activation).

– (Self)ref 77 = Sci. China. Chem. (2021) <https://doi.org/10.1007/s11426-021-1035-5>:

This paper entitled “Benzylic Aroylation of Toluenes with Unactivated Tertiary Benzamides Promoted by Directed ortho-Lithiation” is an original research article submitted by the authors in late-March 2021; I am a little surprised that this work was not referred to in the initial submission to COMMSCEM; as it is very similar indeed. Thus, I would like to raise my concern with respect to the originality of the present submission to COMMSCEM. The proposed concept of C–C bond

formation between tertiary benzamides (electrophile) and weakly C–H acidic substrates (pro-nucleophile) –promoted by a sterically demanding strong base (LDA; generating a secondary amine as conjugate acid [proton source] and thus potentially an equilibrium)– is essentially identical (cf. Scheme 1d in ref 77 vs. Figure 1d in the present submission). Likewise, the suggested reaction mechanism (“indirect” deprotonation of pro-nucleophile via directed ortho-lithiation of the tertiary benzamide electrophile) –one of the stronger and more interesting/original points of the present submission– is identical in both cases (cf. Scheme 5 in ref 77 vs. Figure 3 in the present submission). In other words, the concept and mechanism of the present submission is an application of the chemistry developed in ref 77: the key difference is that in ref 77 toluenes ($pK_a \sim 43$) were used as weakly C–H acidic pro-nucleophiles, whereas in the present submission methyl sulfides ($pK_a \sim 38$) were employed. That said, the present submission still represents a challenging transformation, and therefore the editors may want to decide whether the degree of novelty is sufficient. If the editors judge it to be OK, then ref 77 must be properly cited in the context of the “mechanism” text portion.

Comment 3.10:

p4/6: the deprotonation/quenching of thioanisole with LDA and LTMP –in the absence of the benzamide– must be carried under the reaction conditions of T1 (e.g. THF, 60 °C; then TMSCl) before stating it is unsuccessful (due to pK_a values that are not comparable anyway; as reported in different solvents). Also, if the bis-ortho-methylated benzamide (not a Brønsted acid) is used as a Lewis basic additive (to enhance the Brønsted basicity of LDA), what would be the outcome when mixing thioanisole and LDA?

Response 3.10:

The pK_a values of HAD, HTMP and thioanisole in THF are 35.7, 37.3 and 38.3, respectively. We also carried out the reaction of 4-phenylthioanisole with LDA and LTMP in THF at 60 °C for 24 h and then quenched by TMSCl. No TMS substituted 4-phenylthioanisole was detected, suggesting the deprotonation reaction of 4-phenylthioanisole by LDA and LTMP failed.

Also, treating 4-phenylthioanisole with LDA in the presence of bis-ortho-methylated benzamide at 60 °C for 24 h followed being quenched by TMSCl gave no TMS substituted 4-phenylthioanisole. More than 95% of the 4-phenylthioanisole was recovered.

I appreciate the efforts; the experimental results do not look very conclusive though. Based on these pK_a values (e.g. 37.3 vs. 38.3), I would expect that in case of LTMP e.g. 10% of thioanisole are deprotonated in the equilibrium, and therefore the silylated product should be obtained in ~10% (NMR) yield [similar for LDA; the pK_a value of benzamide is 37.8 so the difference to thioanisole (38.3) is fairly small]. It is not because the silylated product was not detected that there was no deprotonation in a first place; it might be an experimental error (not fully anhydrous conditions and small-scale experiment). Anyway, I don't expect further experiments, but it is better to include a footnote with an appropriate text regarding the attempted deprotonation/silylation (somewhere in the mechanism portion).

Comment 3.11:

p4: there should be also “steric hindrance” included in the base/nucleophile discussion (i.e., bulky amides vs. slim alkyl reagents); it is completely missing.

Response 3.11:

Thanks for your kind suggestion. The steric hindrance, the basicity and the nucleophilicity and their relationship is interesting. We added a comment in the discussion about table 1 entries 3,4 VS 5,6.

OK, addressed; see also the annotated manuscript.

Comment 3.12:

p4/6: are tertiary alcohols obtained at any stage in any of the conducted experiments (particularly for the alkyl lithium reagents and/or in the final control experiment)?

Response 3.12:

The reaction between benzamide and methyl sulfides using n-BuLi as base give tertiary alcohols as by-products. Notes about the tertiary alcohols were added in Table 1, Table S2. A detail explanation about the by-products was added in Figure S1 in SI.

OK, addressed; see also the annotated manuscript.

Comment 3.13:

p4: regarding the amount of LDA used, refer to the SI (TS-1).

Response 3.13:

The amount of LDA was demonstrated clearly in Table S1 in the revised SI. The comment was referred to Table S1.

OK, addressed.

Comment 3.14:

p4: in a similar context, clarify the following points:

What is the outcome if Me-SPh (\rightarrow stoichiometric LDA; secondary center created) is replaced:

(a) by Et-SPh or TMSCH₂-SPh (\rightarrow stoichiometric LDA; tertiary center created) ?

(b) by iPr-SPh (\rightarrow potentially catalytic LDA; quaternary center created) ?

[may require more harsh conditions; dioxane, 80+ ° C]

Response 3.14:

We tried the reaction between Et-SPh or iPr-SPh and tertiary benzamide in 1,4-dioxane under 80 ° C and no desired benzoylated product was detected. In the both cases, the tertiary benzamide was recovered with more than 90% recovery. The reaction displays excellent selectivity to methyl C-H bond possibly because of the four-membered ring transition state structure. The steric hindrance prevents the formation of the highly charged transition state.

OK, addressed.

Comment 3.15:

p4/5 (T2/3): regarding the scope (which is in principle very good), what is the outcome in case of methyl sulfides or tertiary amides bearing an Ar group with an EWG (there is not any example in both Tables)?

Response 3.15:

We tried the reactions between tertiary amide with methyl sulfides bearing an Ar group with 4-CN, 4-NO₂ and 4-CO₂Me, respectively. However, none benzoylated products were detected and tertiary amide were recovered with more than 90% recovery in the three cases. We believe that the above failed examples were due to the incompatibility between the EWGs and LDA in our reaction system. For tertiary amide, those bearing 4-CN, 4-Bz, 4-F and 3-F (not bearing an Ar with 4-CN, 4-Bz, 4-F and 3-F) on the phenyl ring also failed to deliver the corresponding products. Besides, we found that 1.1 equiv. of tert-butyl benzoate used as additive could inhibit the reaction between tertiary amide and thioanisole promoted by LDA in THF under 40 ° C dramatically. We guessed that additional coordination group would possibly catch the LDA and prevent the following process.

OK, addressed. May warrant a footnote in the scope section ?

Comment 3.16:

p4/5 (T2/3): adjust structural display of "tBu" and "iPr" in the Tables.

Response 3.16:

The structural display of "tBu" and "iPr" in the Table 2 and Table 3 have been adjusted.

OK, addressed.

Comment 3.17:

Amide scope: exclusive use of aromatic tertiary N-isopropyl benzamides (= drawback of this methodology): what is the scope of the NR₂ portion? For instance, how about NEt₂, N(TMS)₂, N(pyrrolidinyl)?

Response 3.17: Please refer to response 1.1.

OK, addressed; refer also to 1.1 and 2.2.

Comment 3.18:

p6/7 (F2/3): the control experiments conducted are reasonable (particularly the deuterium labelling ones) and support the proposed mechanistic pathway. It would be good though to try detecting postulated reaction intermediates (e.g. A and D) by mild HRMS methods (obviously in the absence of the sulfide and water, respectively).

Response 3.18:

For the pK_a values of HAD and thioanisole in THF are 35.7 and 38.3, it is not easy to find A in the deprotonation equilibrium. We synthesized the ortho-lithiation intermediate according to the reported method (Angew. Chem. Int. Ed. 2001, 40, 1238). The complex exists in its dimeric structure [A]₂. Treating [A]₂ with 2.8 equiv. of thioanisole in THF at 25 °C delivered benzoylated product 3aa with almost quantitative yield (98%) (SI, 3.7).

The quench of the reaction with BnBr gave further α-benzylated product 6a. These control experiments could be regarded as the evidence of intermediate A and D.

I assume the authors mean benzamide (pK_a = 37.8) not thioanisole (pK_a = 38.3), right ? Well, the experiment could have been done with benzamide and LTMP instead of LDA (pK_a [HTMP] = 37.3) – then there would have been a reasonable chance for detection of A. Anyway, I don't expect this experiment to be done now; the control experiment based on Clayden's procedure (ACIE 2001) is convincing. Fixed.

Comment 3.19:

Sulfide deprotonation: include a suitable ref of a relevant Ar–Li-assisted "σ-bond metathesis-type" C–H bond metalation.

Response 3.19:

Please refer to response 2.3

OK, addressed; refer also to 2.3.

Comment 3.20:

TOC: might be more reader-friendly to move "Control Experiments" forward (S–1).

Response 3.20:

“Control Experiments” was moved forward to the third part in SI.

OK, addressed.

Comment 3.21:

General Information: clarify the LDA confusion; add information on mp measurement (S–2).

Response 3.21:

For the LDA confusion, refer to response 1.7. An information about the mp measurement was added at the end of general information.

OK, addressed; refer also to 1.7.

Comment 3.22:

General Procedures: better to add “m” [in mg] of methyl sulfide reagents, not just show “V” [in μL] (S–4, S–26).

Response 3.22:

The mass information was added into the general procedures and Table S1.

OK, addressed.

Comment 3.23:

Analytical Data: there seem to be accuracy issues with NMR data (almost on every single page ... S–6 ~ S–26); must be addressed; use the term “colorless” (not “white”) solid; some mp’s look strange in terms of significant numbers (S–10, S–11, S–22). NMR Spectra: these display indeed sufficient purity of the isolated products; one exception though: based on the ^1H NMR chart, product 3ea seems not pure (1.2~1.3 ppm); better to repurify and up-date yield.

Response 3.23:

Many thanks for your kind careful review. We corrected the mistakes as pointed out. Compound 3ea was repurified and the yield was up-dated. We have rechecked the ^{13}C NMR data for all of the acquired compounds. For reported compounds, we have compared the spectra acquired with the literature spectra. For example, in the ^{13}C NMR spectra of compounds 3aj and 5ah showed in the literatures, the peak numbers didn’t consist with the structures either. This is mainly due to the overlap of peaks for the very closed chemical shifts.

OK, addressed; refer also to 1.8.

Comment 3.24:

Reaction Optimization: clarify the LDA point; comment on the possibility of tertiary alcohol formation through nucleophilic addition in case of alkyl lithium species (S–3).

Response 3.24:

For the LDA point, see response 1.7; for the tertiary alcohol formation, see response 3.11.

OK, addressed; refer also to 1.7 and 3.11.

Comment 3.25:

Control experiments: clarify two points (S–28, S–30).

Response 3.25:

In S-28, 3a-d was obtained with 11% deuterium incorporation. No tertiary alcohol was detected in the reaction of ketone 3aa with thioanisole 2a-d in the presence of LDA.

OK, addressed.

Overall: The results in the present submission are in principle novel, and the manuscript provides strong evidence for its conclusions; the data being technically sound (with a few citation issues to be addressed though). The chemistry developed here is important in the context of organic synthesis and more specifically C–H bond activation (Organic/Inorganic Chemistry). Strictly conceptually speaking though, there is precedence by the same authors (ref 77), which includes the demonstrated mechanism (the key difference being the switch to another weakly C–H acidic proton nucleophile class). As this reaction methodology corresponds still to a challenging transformation, the editors may want to take the final decision.

Reviewer #4 (Remarks to the Author):

The question is whether we consider a N,N-diisopropyl benzamide to be activated? This is a bit of a 'how long is a piece of string' question - it is not as activated as e.g. N-Boc or N-Ts protected amides as the authors point out but it's certainly more activated than the parent benzamide (due to steric bulk affecting C-N rotation or DMA (due to the arene), etc..

Can I suggest replacing 'unactivated benzamides' with a more neutral term that allows the reader to place these on the 'scale' of amide reactivity, such as 'N,N-dialkyl benzamides'? This makes it clear that the reaction works on less activated benzamides (compared to N-Boc, N-Ts, Weinreb, etc.) but that there is some necessary substitution there. I note that N,N-dimethyl and N,N-diethyl, etc. don't give yields that are as high, so I think there is some bulk needed on the nitrogen and therefore some degree of activation.

Dear reviewers,

I would like to thank you and all the reviewers for the valuable comments and suggestions that have undoubtedly improved the quality of the manuscript. I really enjoyed reading the comments and benefited greatly.

First of all, I would like to explain the “parallel submission” issue because it is about my reputation and the novelty of this study. We originally submitted this manuscript to Nature Communication at Nov. 11th, 2020 (NCOMMS-20-45217). We appealed once and received the final decline decision on Mar. 5th. The editor suggested us to consider transferring the manuscript to Communications Chemistry. We accepted this suggestion and transferred the manuscript to your journal on Mar. 11th simply by clicking a transfer link. Communications Chemistry is a new journal and we did not know how long time it would take. The student urgently needed a paper for his doctoral degree defense. So, we submitted the work about toluene to Science China Chemistry on Mar. 22th and received a revision decision on Apr. 15th. We are happy to soon receive the decision from you on Apr. 23th. During the reversion of this manuscript, the work about toluene was accepted on May 28th and we added it as a reference in the revised manuscript. We did not mean to submitted two papers in close time. Despite that, we are sorry that we didn't inform you and the referees all the details earlier. We agree that the two work with toluene and methyl sulfides share some similarity in the reaction process, in particular the mechanism. Actually, we are happy and proud to find this because the similarly demonstrated that the reaction is a general process rather than a special exception. The toluenes and methyl sulfinds are totally two different kinds of substrate, they don't impact their novelty relatively.

In light of the received comments, we would like to present a revised manuscript. In this revised version, we tuned our tone about “unactivated benzamide”, cited our recent paper about toluene, adjusted some tables in SI and tried to revise the formatting to make it better suit the journal. We deleted some references and added some new ones along with some comments in the text. Most parts changed were highlighted in yellow.

Best regards,

Sincerely yours,

Bing-Tao Guan

Reviewer 1:

Comment 1-1:

“Unactivated” should be removed from the title and from throughout the manuscript.

Response #1-1:

“Unactivated benzamides” was changed to “N,N-dialkyl benzamides” throughout the manuscript.

Comment 1-2:

Ref. 33 should be deleted because it involve N,N-Di-Boc of primary amides. It is too much to be considered as amides.

Response #1-2:

Reference 33 was deleted in the revised manuscript.

Comment 1-3:

The response for not alternate the abstract in not convincing. It should be revised to approach the reality and common sense.

Response 1-3:

We revised the title and abstract as suggesting by the editor.

Comment 1-4:

Please re-cite the following two deleted references (ref. 36 and 41 in the previous version) because they involve amides:

36. Huang, P. & Chen, H. Ni-Catalyzed cross-coupling reactions of N-acylpyrrole-type amides with organoboron reagents. Chem. Commun. 53, 12584-12587 (2017).

41. Meng, G., Szostak, R. & Szostak, M. Suzuki–Miyaura Cross-Coupling of N-Acylpyrroles and Pyrazoles: Planar, Electronically Activated Amides in Catalytic N–C Cleavage. Org. Lett. 19, 3596-3599 (2017).

Response 1-4:

We re-cited the two deleted references as reference 33 and 34 in the revised manuscript.

Comment 1-5:

In the text, “Huang and Charette” should be revised as “Charette and Huang”.

Response 1-5:

“Huang and Charette” was revised as “Charette and Huang”.

Comment 1-6:

*In the supporting information, please check the ¹³C NMR data to ensure that the carbon numbers are in consist with the molecular formula (e.g. **3al**).*

Response 1-6:

We have carefully re-checked the ¹³C NMR data of **3al**. There are 13 peaks that is less than the theoretical value. The overlap of carbon signal accounts for the apparent less of carbon numbers.

Comment 1-7:

*In the supporting information, please check the ¹³C NMR data to ensure the correctness (e.g. **3am**):*

it lacks the data for two Me).

Response 1-7:

We have re-checked the ^{13}C NMR data of **3am**. The peak of the methyl group is δ 24.0. There are several overlap of the aromatic carbons peaks.

Comment 1-8:

Please correct the style for all IUPAC names of compounds: The first letter in capital style, "tert" in italic style, etc.

Response 1-8:

We have corrected all the style for all IUPAC names of compounds.

Comment 1-9:

In the section: "3.2 A general procedure for the alkylation of benzamide with *i*-PrSMe": how is possible: "The aqueous solution was dried over anhydrous Na_2SO_4 ."? Please check all.

Response 1-9:

The expression was revised to "The resulting mixture was dried over anhydrous Na_2SO_4 ."

Comment 1-10:

In the section: "4.3 A gram-scale process for the alkylation of benzamide with thioanisole": "The resulting mixture was extracted thrice with"?

Response 1-10:

The expression was revised to "After separation, the aqueous solution was extracted thrice with...".

Reviewer 3:

Comment 3-1:

However, please indicate in Table S3 –for the cases with 0–69% NMR yield– what the "other material" is: unreacted amide **1** and/or (un)defined side-products?

Response 3-1:

Thanks for the kind comments. The reaction was promoted by the *ortho*-lithiation (please see Fig. 3 A Plausible Reaction Pathway). The *ortho*-lithiated amide could also act as nucleophile and underwent acylation with another molecular *N,N*-dialkylamide, affording *ortho*-amide substituted benzophenones. For example, while using *N,N*-diethylamide and *N*-benzoyl piperidine as amides in the alkylation reaction, we could obtain the corresponding benzophenones with 63% and 68% yield, respectively. For other cases with 0-69% NMR yields, most benzamides were consumed. Some of them afforded the by-product and some just led to more complex undefined by-products. These results were summarized in table S4 with two notes.

Comment 3-4:

The term "weak" should not appear at all in this context – it is misleading and inaccurate. For LDA/LTMP, please use the term "strong" base, or alternatively "moderately strong" base compared with alkyl lithium reagents (in the context of weakly acidic C–H bonds).

Response 3-4:

The strength of alkalinity is a relative concept. When LDA fails to deprotonate methyl sulfide, it is a weaker base than the carbanion. So, we don't think there is any problem to call LDA as a "relatively weak base". But considering that most people think of bases about the water system, we delete the comment "relatively weak base" about LDA to avoid possible misunderstanding and misleading. We tuned our tone about the LDA to "a readily available base" or "a relatively weaker base than lithium alkyls".

Comment 3-5:

OK, addressed; refer also to 3.19. Please add the refs mentioned above (thioanisole deprotonation by PhLi) in the mechanism discussion text portion.

Response 3-5:

Thanks for the comments and suggestion. The deprotonation of sulfides with strong bases was already discuss in the text (ref. 59-63). The strong bases include n-BuLi, t-BuLi, n-BuMgBr and PhLi. PhLi actually was rarely used as the base for thioanisole deprotonation and did not display any better reactivity than n-BuLi. We didn't find any advantage or difference of PhLi in the deprotonation reaction. So, we don't think it is necessary to emphasize the reactivity in the mechanism discussion.

Comment 3-6:

OK, addressed. However, please correct the term (decreased nucleophilicity) as well in Figure 1a (see annotated manuscript).

Response 3-6:

Thanks for pointing out this mistake. We change "decreased nucleophilicity" to "electrophilicity nucleophilicity" in Figure 1a.

Comment 3-7:

It would have been better to include (the part on tertiary benzamides that react with in situ-generated alkyl anions to form the corresponding ketones) for a more scholarly presentation (as it is very much related). If it is a matter of space, you could have removed Figure 1a (knowledge of undergrad level year 1). Anyway, I won't insist – the editor can decide.

Response 3-7:

Thanks for the comments and understanding. We agree that the reaction of tertiary amides in Liu's work is very much related with this topic. But the most interesting part of Liu's work is the "chemodivergent transformation of various amides". The emphasis on the tertiary amides with a visual picture would lose the whole picture and possibly lead to a one-sided impression. Thus, we would still prefer to appreciate Liu's excellent work and give a comment in the text.

Comment 3-8:

I appreciate the authors' standpoint, but I disagree. I had stated that self-citation (to a reasonable extent) is of course OK. However, relevant publications by other groups –particularly when these were published earlier and displaying alkali metal amide catalysis rather than the "stoichiometric" use of alkali metal amides, like in the present submission– must be respected and thus included. By the way, when the pKa difference (base vs. C–H acidic substrate) is e.g. /– 3 units, a deprotonative

equilibrium is expected – it does not have to be explicitly stated. Therefore, the following [adjusted] citations are expected (for accuracy/fairness – it does not take anything away from the present submission, but the reader must be more suitably informed):

(1) Initial (self)refs 81 and 83 correspond now to (self)refs 74 (ACIE 2018, 57, 1650) and 75 (OL 2019, 21, 5351); these are appropriate indeed (no issue at all). It is appreciated that the authors removed –on their own– initial self-ref 82 (ACIE 2018, 57, 8245): a beautiful study but not so relevant in the context of the present submission.

(2) The following (earlier) papers by other groups must be cited:

– JACS 2017, 139, 4362:

Published earlier than refs 74 and 75; catalytic use of Li/Na/K amide (Na = best) as a “moderate base” for the deprotonation of weakly acidic C–H bonds; give generally better results than MeLi (“strong base”; see Table 1); an equilibrium (deprotonation/reprotonation; due to the generation of a non-gaseous conjugate acid = secondary amine) has been clearly proposed (display of equilibrium arrow in Scheme 1b) and proved in both optimization experiments (Table 1: formation of isomerization side-product 1’a; plus text) and mechanism experiments (Scheme 4i: formation of isomerization side-product 1’a; plus text).

– ACIE 2018, 57, 6896:

Published earlier than ref 75; catalytic use of K amide (in the presence of LiOR) as a “moderate base” for the deprotonation of weakly acidic C–H bonds. Here, a deprotonative equilibrium is simply expected because of the small pKa difference – no need to mention.

– NatComm 2018, 9, 3365:

Published earlier than ref 75; stoichiometric use of Na/K amide (Na = best; in the presence of CsX) as a “moderate base” for the deprotonation of weakly acidic C–H bonds. Here, a deprotonative equilibrium has been clearly stated (display of equilibrium arrow between 2 and C in Figure 3; plus text).

In turn, may I ask to please add this refs ?

(3) The authors have included two new (self)refs 76 & 77 (and I am a little surprised that these have not been drawn clearly to the attention of the editors/reviewers):

– (Self)ref 76 = *Sci. China. Chem.* 51, 201 (2021):

This paper is a kind of account-style review by the authors. Of course, it is OK to be included as it refers to earlier contributions by the authors and other PIs (in this specific field of C–H bond activation).

– (Self)ref 77 = *Sci. China. Chem.* (2021) <https://doi.org/10.1007/s11426-021-1035-5>:

This paper entitled “Benzylic Aroylation of Toluenes with Unactivated Tertiary Benzamides Promoted by Directed ortho-Lithiation” is an original research article submitted by the authors in late-March 2021; I am a little surprised that this work was not referred to in the initial submission to COMMSCHEM; as it is very similar indeed. Thus, I would like to raise my concern with respect to the originality of the present submission to COMMSCHEM. The proposed concept of C–C bond formation between tertiary benzamides (electrophile) and weakly C–H acidic substrates (pro-nucleophile) –promoted by a sterically demanding strong base (LDA; generating a secondary amine as conjugate acid [proton source] and thus potentially an equilibrium)– is essentially identical (cf. Scheme 1d in ref 77 vs. Figure 1d in the present submission). Likewise, the suggested reaction mechanism (“indirect” deprotonation of pro-nucleophile via directed ortho-lithiation of the tertiary benzamide electrophile) –one of the stronger and more interesting/original points of

the present submission– is identical in both cases (cf. Scheme 5 in ref 77 vs. Figure 3 in the present submission). In other words, the concept and mechanism of the present submission is an application of the chemistry developed in ref 77: the key difference is that in ref 77 toluenes ($pK_a \sim 43$) were used as weakly C–H acidic pro-nucleophiles, whereas in the present submission methyl sulfides ($pK_a \sim 38$) were employed. That said, the present submission still represents a challenging transformation, and therefore the editors may want to decide whether the degree of novelty is sufficient. If the editors judge it to be OK, then ref 77 must be properly cited in the context of the “mechanism” text portion.

Response 3-8:

Thanks for the kind comments and detail explanation. We earlier just wanted to introduce the logic basis of the present work and didn't pay much attention to the background about the base catalyzed reactions. We are now convinced to cite the pointed papers (JACS 2017, 139, 4362; ACIE 2018, 57, 6896; NatComm 2018, 9, 3365) as refs 78-80 in the revised manuscript. For the new ref. 76 and 77 (now ref. 77 and 81) and also the “parallel submissions” and novelty of this work, I provided an explanation at very beginning of this response letter. In addition, a comment and a citation of ref. 81 were added in the text.

Comment 3-9:

I appreciate the efforts; the experimental results do not look very conclusive though. Based on these pK_a values (e.g. 37.3 vs. 38.3), I would expect that in case of LTMP e.g. 10% of thioanisole are deprotonated in the equilibrium, and therefore the silylated product should be obtained in ~10% (NMR) yield [similar for LDA; the pK_a value of benzamide is 37.8 so the difference to thioanisole (38.3) is fairly small]. It is not because the silylated product was not detected that there was no deprotonation in a first place; it might be an experimental error (not fully anhydrous conditions and small-scale experiment). Anyway, I don't expect further experiments, but it is better to include a footnote with an appropriate text regarding the attempted deprotonation/silylation (somewhere in the mechanism portion).

Response 3-9:

Thanks for the comments. We do agree with the referee about the deprotonation of thioanisole with LiTMP (10% of thioanisole are deprotonated in the equilibrium). This conclusion comes from the pK_a value measured by Fraser (Fraser, R.-R.; Bresse, M.; Mansour, T.-S. pK_a Measurements in tetrahydrofuran. J. Chem. Soc., Chem. Commun. 1983, 620.). The pK_a values of have been assigned with reference to 2-methyl-1,3-dithiane, whose pK_a was assigned as 37.8. Thus, he didn't directly carry out the reaction between thioanisole and LiTMP. In addition, we carried out the reaction with 4-phenylthioanisole. There might be an experimental error. It is also possible that we did not make the mistake. It seems to be complex. We would like to figure it out later. But for now, once we have proved that the benzamide participates in the deprotonation process, the direct deprotonation of anisole with lithium amide is not a crucial problem.

Comment 3-10:

p4/5 (T2/3): regarding the scope (which is in principle very good), what is the outcome in case of methyl sulfides or tertiary amides bearing an Ar group with an EWG (there is not any example in both Tables)? Response by authors: We tried the reactions between tertiary amide with methyl sulfides bearing an Ar group with 4-CN, 4-NO₂ and 4-CO₂Me, respectively. However, none

benzoylated products were detected and tertiary amide were recovered with more than 90% recovery in the three cases. We believe that the above failed examples were due to the incompatibility between the EWGs and LDA in our reaction system. For tertiary amide, those bearing 4-CN, 4-Bz, 4-F and 3-F (not bearing an Ar with 4-CN, 4-Bz, 4-F and 3-F) on the phenyl ring also failed to deliver the corresponding products. Besides, we found that 1.1 equiv. of tert-butyl benzoate used as additive could inhibit the reaction between tertiary amide and thioanisole promoted by LDA in THF under 40 °C dramatically. We guessed that additional coordination group would possibly catch the LDA and prevent the following process. Comments by reviewer: OK, addressed. May warrant a footnote in the scope section?

Response 3-10:

Thanks for the comments. We summarized the results in table S4 and gave a comment below.

Comment 3-II:

I assume the authors mean benzamide ($pK_a = 37.8$) not thioanisole ($pK_a = 38.3$), right?

Response 3-11:

Yes, we mean benzamide ($pK_a = 37.8$).

Reviewer 4

Comment 4-I:

The question is whether we consider a *N,N*-diisopropyl benzamide to be activated? This is a bit of a 'how long is a piece of string' question - it is not as activated as e.g. *N*-Boc or *N*-Ts protected amides as the authors point out but it's certainly more activated than the parent benzamide (due to steric bulk affecting C-N rotation) or DMA (due to the arene), etc..

Can I suggest replacing 'unactivated benzamides' with a more neutral term that allows the reader to place these on the 'scale' of amide reactivity, such as '*N,N*-dialkyl benzamides'? This makes it clear that the reaction works on less activated benzamides (compared to *N*-Boc, *N*-Ts, Weinreb, etc.) but that there is some necessary substitution there. I note that *N,N*-dimethyl and *N,N*-diethyl, etc. don't give yields that are as high, so I think there is some bulk needed on the nitrogen and therefore some degree of activation.

Response 4-1:

Thanks for the kind comments. We used "*N,N*-dialkyl benzamides" instead of "unactivated benzamides" in the revised manuscript.